# EfficientRefiner: An Efficient Refinement Method for Macro Placements Generated by Off-the-Shelf Placers

## Abstract

A refinement stage on macro placements generated by SOTA off-the-shelf placers can further improve the layout quality, as this stage compensates for the sub-optimality arising from lack of full-layout awareness in RL-based placers, as well as the quality degradation resulting from the overlap-resolving legalization step in analytical placers. Nevertheless, existing RL-based refinement techniques often incur high computational cost. This paper proposes EfficientRefiner, which leverages an efficient analytical framework to refine macro layouts produced by existing placement approaches, achieving reduced computational overhead while improving layout quality. EfficientRefiner encodes macro positions as learnable vectors and optimizes an objective function that integrates both target metrics and placement constraints via gradient descent. It introduces a novel fine-grained pairwise overlap formulation tailored for macro refinement, which overcomes the limitations of prior density-based objectives in analytical methods by effectively minimizing overlaps without inducing excessive spreading that could degrade layout quality. Moreover, EfficientRefiner enhances efficiency and scalability through pruning algorithms and GPU acceleration. Experimental results show that, when considering both HPWL and regularity metrics for optimization, it improves average HPWL by **7.20%–34.71% within 10 minutes** on the ISPD2005 benchmark, and achieves average timing gains of **20% WNS and 29% TNS** on PPA-supported ChiPBench circuits.

## 1 Introduction

Chip placement is a critical stage in Electronic Design Automation (EDA), as it strongly influences subsequent steps such as clock tree synthesis and routing, and significantly affects the overall quality of the chip design. The goal is to generate an optimized layout for both large functional modules (macros) and small logic gates (standard cells), ensuring that no overlaps occur while improving key objectives (e.g., proxy wirelength metric, and final Power, Performance and Area (PPA) metrics). Macro placement plays a decisive role within the overall placement task, as macros are much larger and more densely connected than standard cells (Geng et al., 2024). Nevertheless, the problem remains highly challenging due to its NP-hard nature and the intricate trade-offs involved in optimizing placement quality under essential design constraints (Wang et al., 2009).

A wide range of approaches have been developed to address the placement problem, with state-of-the-art methods mainly falling into analytical-based and Reinforcement Learning (RL)-based categories. Analytical methods (Lin et al., 2019; Lu et al., 2015; Cheng et al., 2018; Chen et al., 2008) formulate differentiable objectives, such as wirelength (capturing the primary optimization goal) and density (encouraging module spreading). Then they optimize these objectives efficiently with gradient-descent. Analytical-based methods leverage global layout information and offer high computational efficiency, but often cause severe macro overlaps which have to be resolved by a subsequent legalization step at the cost of significant performance degradation (Lai et al., 2022). RL-based methods (Mirhoseini et al., 2021; Lai et al., 2022; 2023; Cheng & Yan, 2021; Geng et al., 2024) formulate placement as a Markov Decision Process (MDP). They mainly learn policies that place modules step by step. These methods have shown promise in generating high-quality placements and are able to avoid overlaps through masking, but they suffer from high computational cost

and limited ability to capture global context. Overall, existing approaches show notable strengths but also leave room for further improvement.

An analytical-based refinement stage has the potential to improve layouts produced by existing placement methods, as it can compensate for the suboptimality caused by incomplete global context information and the quality loss from legalization. Previous work such as MaskRegulate (Xue et al., 2024) has explored RL to refine DreamPlace-generated layouts, but this approach requires dataset-specific training to achieve the best results and updates only one module per iteration, which is inefficient. In contrast, we view refinement as a post-processing stage that should impose minimal additional runtime, and thus adopt an efficient analytical framework to implement refinement and explore its effectiveness in enhancing placement quality.

Effectively handling overlaps is particularly critical when applying analytical methods for macro refinement, because (1) macros are large, vary greatly in size, and densely connected, which increases the likelihood of overlaps and makes legalization more likely to degrade the refined layout; (2) large macro perturbations at legalization can substantially diminish the value of refinement on layouts already with high quality (Lai et al., 2022). Existing analytical methods generally address overlaps using coarse-grained density functions, which partition the placement region into grids and drive each grid toward a target density. However, this strategy faces two major issues for handling macro overlaps. One issue is that minimizing grid-based objectives does not guarantee complete removal, often leaving significant overlaps among macros unresolved. The other is that many methods rely on repulsive forces between modules, but in high-density regions these forces may continue acting even after modules have moved away, leading to unnecessary spreading that may hinder effective optimization of key objectives (Cheng et al., 2018).

To tackle the above issues, we introduce a fine-grained, module-pair-based overlap function that effectively reduces macro overlaps. This function explicitly computes overlaps between every pair of macros and aggregates them to obtain the total overlap, providing a more accurate representation than grid-based formulations. To address the higher computational cost of fine-grained modeling, we employ algorithmic optimizations together with GPU acceleration for computation of both the overlap function and its gradient, resulting in substantial efficiency gains and improved scalability. Building on this novel overlap formulation and its efficient implementation, we develop EfficientRefiner, a layout refinement method specifically suitable for efficient macro refinement.

EfficientRefiner can seamlessly integrate with any placement approach to optimize placement objectives while maintaining low overlap. The main contributions are as follows: (1) We introduce a novel module-pair-based overlap function tailored for refinement scenarios, which provides a more accurate representation of overlaps and enables effective overlap reduction. (2) We design an efficient pruning scheme for overlap computation across large numbers of modules, combined with a GPU-accelerated refinement implementation, to ensure efficiency and scalability. (3) We incorporate multiple optimization objectives in our experiments, including HPWL and the regularity metric to improve PPA. Experimental results show that our approach improves average HPWL by 7.20%–34.71% on the ISPD2005 benchmark, and improves WNS and TNS by 20% and 29%, respectively, on PPA-supported ChiPBench circuits.

## 2 RELATED WORK

We begin by reviewing existing placement methods, whose outputs provide the initial layouts for our refinement. We then discuss prior refinement approaches that further improve placement quality.

### 2.1 PLACEMENT METHODS

Placement methods can be broadly categorized into constructive and iterative adjustment methods (Shahookar & Mazumder, 1991). **Constructive methods** start from an empty placement region and generate layouts from scratch. Early work is mainly partition-based (Breuer, 1977; Agnihotri et al., 2003; Can Yildiz & Madden, 2001; Khatkhate et al., 2004), where modules are clustered using min-cut algorithms (Fiduccia & Mattheyses, 1988; Karypis et al., 1997; Alpert et al., 1997) and assigned to subregions in a recursive divide-and-conquer manner until clusters reach a manageable size. Recent work (Mirhoseini et al., 2021; Cheng & Yan, 2021; Lai et al., 2022; 2023; Geng et al., 2024) leverages the strong learning capability of RL to achieve state-of-the-art results. These methods

train RL agents to construct layouts by sequentially placing modules. MaskPlace (Lai et al., 2022) introduces masks that encode layout occupancy and wirelength increments to guide optimization. This mechanism effectively removes overlaps and significantly improves macro placement quality, and thus has been widely adopted in subsequent studies (Geng et al., 2024; Shi et al., 2023; Geng et al., 2025). ChipFormer (Lai et al., 2023) improves the efficiency of RL methods by combining offline training with online fine-tuning. Although many constructive approaches achieve strong performance, they lack foresight of the global layout to guide optimization. And among them, RL-based methods, although often the most effective, require expensive training and face difficulties scaling to placements with a large number of modules.

**Iterative adjustment methods** start from relatively poor initial layouts (e.g., random initialization) and make iterative improvement. Stochastic-based adjustment methods, such as simulated annealing (Sechen & Sangiovanni-Vincentelli, 1985; Adya & Markov, 2001; Ho et al., 2004; Shunmugatham-mal et al., 2020; Yang et al., 2000) or evolutionary algorithms (Shi et al., 2023), improve layouts through numerous adjustments. These methods often require repeatedly executing a time-consuming process, which maps genotype solutions, which are convenient for adjustment (Chang et al., 2000; Hong et al., 2000; Murata et al., 1996), to phenotype solutions for evaluation. Besides, LaMPlace (Geng et al., 2025), adopts the WireMask-BBO framework but guides optimization with PPA-related masks to improve ultimate placement metrics. Analytical-based adjustment methods (Lin et al., 2019; Lu et al., 2015; Cheng et al., 2018; Chen et al., 2008; Spindler et al., 2008; Sigl et al., 1991; Viswanathan et al., 2007; Kahng et al., 2005) are highly efficient. They model placement objectives (e.g., wirelength) and constraints (e.g., density) as differentiable functions of module coordinates and optimize them using gradient-based techniques. However, the density formulation, intended to encourage roughly uniform module distribution, is ineffective at fully eliminating macro overlaps. This often results in substantial macro overlaps that must be resolved during the legalization stage, which can in turn significantly alter the layout and degrade overall placement quality.

## 2.2 REFINEMENT METHODS

The above placement methods still leave room for improvement, which can be addressed through an additional refinement process. Existing methods leverage RL to adjust layouts produced by Dream-Place. MaskRegulate (Xue et al., 2024) learns an adjustment policy that relocates one macro per step guided by masks similar to MaskPlace. Chiang et al. (2025) trains a deep Q-network to adjust groups of blocks (i.e., macros and standard cell clusters) simultaneously at each step, generating mixed-size placement prototypes for subsequent DreamPlace optimization. However, reinforcement learning approaches are computationally expensive and can only adjust a limited number of modules per iteration. To reduce the overhead of the post-processing refinement stage, we explore an analytical framework for refinement and introduce a fine-grained overlap function to address the limitations of analytical methods in handling macro overlaps.

## 3 PRELIMINARIES AND NOTATIONS

The goal of macro placement is to determine the optimal arrangement of macros within a rectangular chip region while ensuring compliance with the non-overlapping constraint. The input includes the width and height of the placement region $(R_w, R_h)$, and a circuit netlist $G(V, E)$ which can be viewed as a hypergraph where modules or ports act as hypernodes, while nets connecting them serve as hyperedges. Modules and ports are connected by nets through pins which serve as I/O interfaces located at fixed positions relative to their corresponding modules.

The optimization objectives include the final Power, Performance, and Area (PPA) metrics and surrogate metrics such as wirelength. In practice, directly evaluating PPA requires time-consuming subsequent steps such as routing. Consequently, surrogate metrics are typically employed. Half-Perimeter Wirelength (HPWL) is a widely used surrogate metric, serving as an estimate of wire-length. A smaller HPWL may indicate reduced routing resource consumption and better performance. Since HPWL is non-differentiable, the weighted-average function (Hsu et al., 2013) is commonly adopted as a differentiable approximation to enable gradient-based optimization. We also employ this differentiable surrogate in our refinement framework. In addition, we incorporate the regularity metric (Xue et al., 2024), which encourages macros to be placed closer to the chip boundary, thereby leaving sufficient space for standard cells and improving both mixed-size placement wirelength and PPA. Detailed definitions of these placement metrics are provided in Appendix A.1.

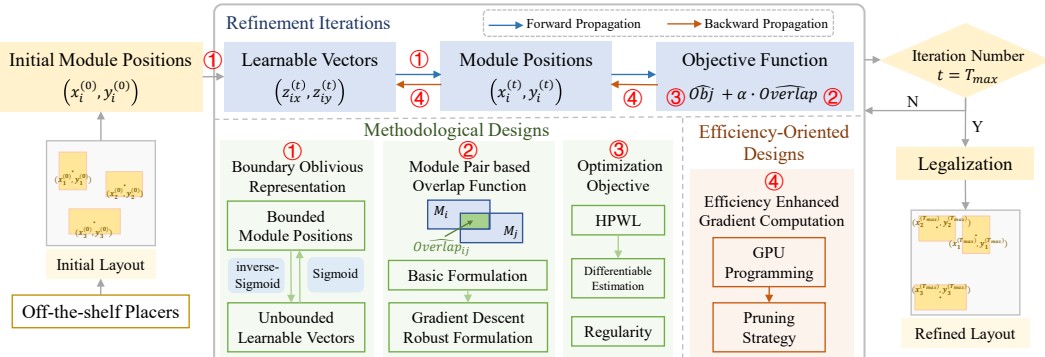

**EfficientRefiner**

Figure 1: **Overview of EfficientRefiner.** Starting from an optimized layout generated by an off-the-shelf placer, EfficientRefiner encodes module positions as unbounded learnable vectors, optimizes a joint objective function consisting of the optimization objective and fine-grained overlap via iterative forward and backward propagation, and applies legalization at the end to resolve any remaining overlaps.

Our refinement task can be described as follows. We receive the initial set of macro positions $S = \{(x_1, y_1), (x_2, y_2), ..., (x_n, y_n)\}$ generated by existing placement methods as input, and aim to find a set $S^{(ref)} = \{(x_1^{(ref)}, y_1^{(ref)}), (x_2^{(ref)}, y_2^{(ref)}), ..., (x_n^{(ref)}, y_n^{(ref)})\}$ of refined locations which satisfies Eq.(1). In the equation, Obj denotes the optimization objective, which can be adapted to different metrics depending on the setting. In this paper we support HPWL and regularity, and may extend them to metrics that more directly reflect PPA in the future.

$$\text{Obj}(S^{(ref)}) < \text{Obj}(S), \quad \text{Overlap}(S^{(ref)}) = 0 \tag{1}$$

We reformulate the refinement problem as an unconstrained optimization, as defined in Eq.(2), by incorporating the overlap constraint into the objective function with a weighting parameter $\alpha$. We employ differentiable approximations during refinement, denoted as $\hat{\text{Obj}}$ and $\hat{\text{Overlap}}$.

$$f = \hat{\text{Obj}} + \alpha \cdot \hat{\text{Overlap}} \tag{2}$$

## 4 EFFICIENTREFINER

Fig.1 presents an overview of EfficientRefiner. EfficientRefiner first represents module positions as learnable vectors and formulate an objective function that integrates placement metrics and constraints (Eq.(2)). Then it refines the layouts by optimizing this objective through iterative gradient descent. Note that with our proposed fine-grained overlap formulation, the overlap remains low throughout the refinement process. After a specified number of iterations, a legalization step is applied to remove any remaining overlaps. This step requires only minor adjustments to module positions and has little impact on the overall layout, as the overlap rate is already low.

Our method consists of several key components. First, the boundary-oblivious module position representation maps module positions to unconstrained learnable vectors to ensure that modules remain within the chip boundary during refinement. Second, the fine-grained, module-pair-based overlap function enables effective overlap removal and reduces the impact of legalization on macro refinement. This is in contrast with coarse-grained grid-based density formulations used in analytical methods, which suffer from two main drawbacks: (1) they often leave overlaps unresolved even when the objective is minimized. (2) Their repulsive force mechanisms continue to push modules after they have moved away from dense regions, causing unnecessary spreading that can degrade performance (Cheng et al., 2018). Third, GPU programming, along with pruning techniques, are employed to enable efficiency and scalability.

### 4.1 BOUNDARY-OBLIVIOUS MODULE POSITION REPRESENTATION

Representing module positions as boundary-oblivious learnable vectors streamlines optimization by eliminating the need to check for boundary violations during refinement. Each bounded module position $(x_i, y_i)$ (constrained within $[\frac{w_i}{2}, R_w - \frac{w_i}{2}] \times [\frac{h_i}{2}, R_h - \frac{h_i}{2}]$) is mapped to an unbounded vector representation $(z_{ix}, z_{iy})$.

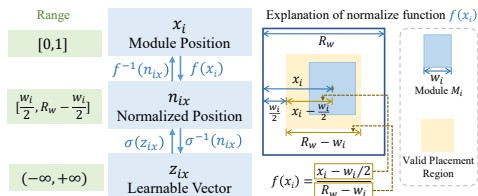

Figure 2: **Boundary-oblivious representation.**

The mapping technique is illustrated in Fig.2 and described in detail below. We take the $x$-direction as an example, as the mapping in the $y$-direction follows similar process. For a module $M_i$ with absolute position $(x_i, y_i)$, the mapping consists of three steps. First, we determine the valid placement region, (i.e., the range that ensures $M_i$ remains within the placement boundary), given by $[w_i/2, R_w - w_i/2]$. Next, we normalize $(x_i, y_i)$ to the interval $[0, 1]$ using the calculated valid region boundary value. Specifically, the normalized $x$-position is computed as the ratio of the distance from the center of $M_i$ to the region's left boundary over the region's width, which is presented as $\frac{x_i - w_i/2}{R_w - w_i}$. Finally, the normalized position is mapped to the learnable vector $z_{ix}$ using the inverse sigmoid function. The complete mapping function is given in Eq.(3). The resulting learnable vector $z_{ix}$ spans the entire real domain and is therefore unbounded.

$$z_{ix} = \sigma^{-1}\left(\frac{x_i - w_i/2}{R_w - w_i}\right) \tag{3}$$

The role of this mapping technique in the overall refinement process is as follows. At the beginning, given an initial layout produced by any placement placement method, the mapping is applied to compute the initial values of the learnable vectors from the current module positions. During each forward propagation, the learnable vectors are then mapped back into module positions through the inverse of Eq.(3), so the objective function can be computed based on these reconstructed positions.

## 4.2 Fine-grained Module Pair based Overlap Function

The fine-grained, module pair-based overlap function is specifically designed for macro refinement and offers several advantages over the density formulations used in analytical methods. First, it ensures more effective overlap reduction, as modules are guaranteed to be non-overlapping when the overlap function reaches its minimum value of zero. Second, it prevents unnecessary module spreading that can degrade placement quality, since the gradient of the overlap function becomes zero once a module no longer overlaps with others.

To formulate the overlap function, we begin with a basic version that aggregates pairwise module overlaps and then extend it to a gradient-descent–robust formulation. This evolution is illustrated in the blue-shaded panel on the middle-left of Fig.1. In the basic version, the gradient vanishes when the overlap reaches its maximum, hindering further adjustment. Therefore, a more robust formulation is introduced to enable effective gradient-based optimization.

**Basic Overlap Formulation.** The basic overlap formulation is defined by aggregating the overlap areas across all module pairs. For a given pair $(M_i, M_j)$, the overlap area $\text{Overlap}_{ij}$ is computed as the product of the overlapping lengths along the $x$- and $y$-directions, as illustrated in Fig.3(a). The exact formulation is provided in Eq.(4).

$$\text{Overlap} = \sum_{M_i, M_j \in V, i \neq j} \text{Overlap}_{ij} = \sum_{M_i, M_j \in V, i \neq j} \text{Overlap}_{ijx} \cdot \text{Overlap}_{ijy} \tag{4}$$

In the above equation, the overlap lengths $\text{Overlap}_{ijx}$ and $\text{Overlap}_{ijy}$ between modules $M_i$ and $M_j$ are defined as follows. Taking the $x$-direction as an example (the $y$-direction is analogous), $\text{Overlap}_{ijx}$ is given by the difference between the minimum of the two right boundaries and the maximum of the two left boundaries when the modules overlap; otherwise, it is zero. The exact formulation is provided in Eq.(5) and illustrated in Fig.3(a).

$$\text{Overlap}_{ijx} = max(0, min(x_i + \frac{w_i}{2}, x_j + \frac{w_j}{2}) - max(x_i - \frac{w_i}{2}, x_j - \frac{w_j}{2})) \tag{5}$$

This basic formulation suffers from a zero-gradient issue that limits its effectiveness in gradient-based optimization for overlap removal. Specifically, when the span of $M_i$ in the $x$-direction is fully contained within that of $M_j$, $\text{Overlap}_{ijx}$ remains fixed and provides no gradient signal. Such cases, as illustrated in Fig.3(b) and 3(c), prevent $M_i$ and $M_j$ from being effectively separated. To address this issue, we revise the formulation and define a gradient-descent-robust version $\hat{\text{Overlap}}_{ijx}$.

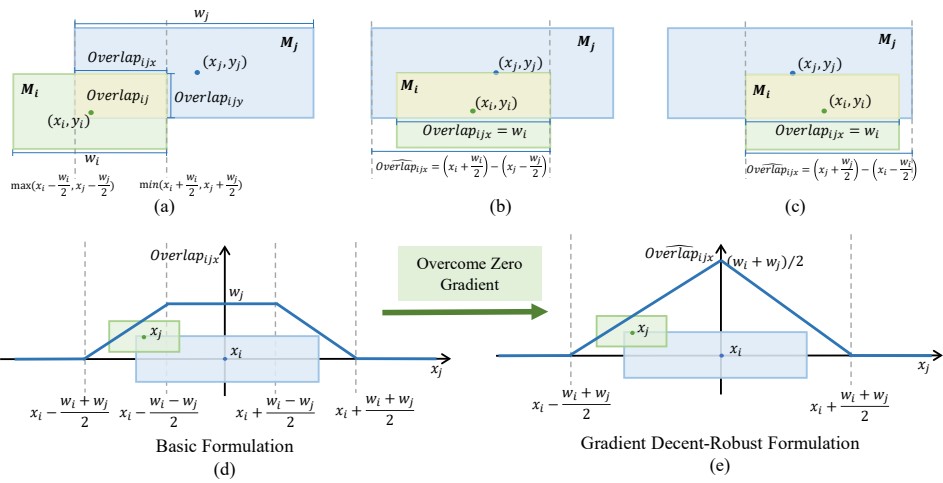

Figure 3: **Basic and Gradient-Descent-Robust Overlap Formulations.** (a) Overlap between two modules is computed as the product of their overlapping lengths along the $x$- and $y$-directions. (b)-(c) Overlapping length in the $x$-direction under the basic and gradient descent robust formulations when module $M_i$ lies to the left of $M_j$ (b) or to the right of $M_j$ (c). (d)-(e) Plots of the $x$-direction overlapping length as a function of $x_j$ under the basic formulation (d) and the gradient descent robust formulation (e).

**Gradient Descent Robust Overlap Formulation.** To resolve the zero-gradient issue, we extend the basic overlap formulation to a gradient-descent-robust version for cases where the span of $M_i$ fully contains that of $M_j$ in either $x$- or $y$-directions. As shown in Fig.3(b) and Fig.3(c), two subcases are considered: (1) if the center of $M_i$ lies to the left of $M_j$, the overlapping length is extended from the right boundary of $M_i$ to the left boundary of $M_j$; (2) if the center of $M_i$ lies to the right, it is extended from the left boundary of $M_i$ to the right boundary of $M_j$. In all other case (i.e., disjoint or partially overlapping spans) $\hat{\text{Overlap}}$ reduces to the basic formulation $\text{Overlap}$. The full formulation is provided in Eq.(6).

$$\hat{\text{Overlap}}_{ijx} = \begin{cases} (x_j + \frac{w_j}{2}) - (x_i - \frac{w_i}{2}), & \text{if } x_j \leq x_i \quad \text{and} \quad |x_i - x_j| < \frac{w_i}{2} + \frac{w_j}{2} \\ (x_i + \frac{w_i}{2}) - (x_j - \frac{w_j}{2}), & \text{if } x_j > x_i \quad \text{and} \quad |x_i - x_j| < \frac{w_i}{2} + \frac{w_j}{2} \\ 0, & \text{otherwise} \end{cases} \quad (6)$$

Figure 3(e) illustrates $\hat{\text{Overlap}}_{ijx}$ as a function of $M_j$'s position with $M_i$ fixed. As $x_j$ moves from $x_i - \frac{w_i + w_j}{2}$ to $x_i + \frac{w_i + w_j}{2}$, the overlap length rises linearly to the peak and then symmetrically decreases to zero. Its derivative maintains an absolute value of 1 within the overlap region, resolving the zero-gradient issue.

## 4.3 EFFICIENCY ENHANCED GRADIENT COMPUTATION

The objective function is optimized using gradient descent as defined in Eq.(7), with $lr$ representing the learning rate.

$$z_{ix} = z_{ix} - lr \cdot \frac{\partial f}{\partial z_{ix}}, \quad z_{ij} = z_{ix} - lr \cdot \frac{\partial f}{\partial z_{ix}} \quad (7)$$

Since the module-pair-based overlap formulation introduces larger gradient computational overhead than previous coarse-grained density formulations, we adopt two acceleration strategies to maintain efficiency and scalability: (1) GPU programming, which leverages the high computational power of GPUs and improves the parallelism of gradient computation; (2) A pruning strategy, which reduces redundant pairwise computations to improve refinement efficiency for large-scale designs.

### 4.3.1 GPU PROGRAMMING

We begin by analyzing the parallelism in gradient computation, which motivates the use of GPU programming to improve efficiency. We then explain the GPU programming scheme in detail.

**Parallelism in Gradient Computation.** The gradient of $f$ with respect to the learnable vector $z_{ix}$ consists of two parts: the derivative of $\hat{\text{Obj}}$ and $\hat{\text{Overlap}}$, respectively, as shown in Eq.(8).

$$\frac{\partial f}{\partial z_{ix}} = \frac{\partial f}{\partial x_i} \cdot \frac{\partial x_i}{\partial z_{ix}} = \left( \frac{\partial \hat{\text{Obj}}}{\partial x_i} + \lambda \cdot \frac{\partial \hat{\text{Overlap}}}{\partial x_i} \right) \cdot \frac{\partial x_i}{\partial z_{ix}} \tag{8}$$

The term $\hat{\text{Overlap}}$ aggregates the contributions of overlaps between all module pairs, and its gradient is given in Eq.(9). We can observe from this formulation that the overlap gradients between module $M_i$ and each other module $M_j$ can be computed independently and then summed, which enables efficient parallelization.

$$\frac{\partial \hat{\text{Overlap}}}{\partial x_i} = \sum_{M_j \in V, j \neq i} \hat{\text{Overlap}}_{ijy} \cdot \frac{\partial \hat{\text{Overlap}}_{ijx}}{\partial x_i} \tag{9}$$

**GPU Programming Scheme.** Based on the parallelism analysis, we implement GPU programming to accelerate computation. Specifically, a dedicated GPU thread is assigned to each module pair to enable parallel computation of the overlap lengths and their corresponding gradients. Then, overlap contributions from all module pairs are accumulated according to Eq.(4) to obtain the overall overlap function, and the gradients are accumulated according to Eq.(9) to yield overlap gradients.

In practice, we adopt the GPU programming interface provided by the Numba library, as it offers greater flexibility in defining GPU threads for parallel computation and better supports the pruning strategy introduced later. In contrast, the more commonly used PyTorch implementation can only compute overlaps between module pairs sequentially under the same space complexity, which leads to significant efficiency degradation, as shown in the experimental section.

### 4.3.2 PRUNING STRATEGY

The pruning strategy further reduces computation when refining a large number of modules by reducing the number of module pairs under consideration. The strategy divides the placement region into rectangular bins, and categorizes modules as either large (with width or height exceeding a bin dimension) or small (fully contained within a bin). For large modules, overlaps and gradients are computed with respect to all other modules in the layout. For small modules, the computation is restricted to pairs formed with modules located in the same bin and its eight neighboring bins. For example, in Fig. 4, module $M_1$ is identified as large and interacts with all other modules, whereas module $M_2$, classified as small, only interacts with modules $M_3$, $M_4$, and $M_5$ residing in its bin and adjacent bins (marked by the yellow shaded region). The detailed algorithm can be find in Appendix A.2.

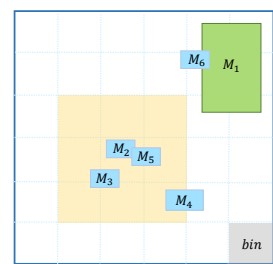

Figure 4: **Example of the pruning strategy.**

## 5 EXPERIMENTS

### 5.1 BENCHMARKS, BASELINES AND SETTINGS

We evaluate the effectiveness of EfficientRefiner on macro and mixed-size HPWL using the widely adopted ISPD2005 (Nam et al., 2005) and ICCAD2015 (Kim et al., 2015) benchmarks, which together contain 16 circuits. We further assess PPA results on 8 circuits from ChiPBench (Wang et al., 2024). We refine multiple state-of-the-art placement methods and compare their results before and after refinement. We also compare against the RL-based method MaskRegulate (Xue et al., 2024) to examine their relative effectiveness in refinement. Additional benchmark statistics, baselines and experimental settings are provided in Appendix A.3.

### 5.2 MAIN RESULTS

We conduct three groups of experiments: (1) optimizing HPWL alone to evaluate macro wirelength reduction; (2) jointly optimizing HPWL and regularity to assess effectiveness on mixed-size placement with respect to wirelength and PPA; and (3) comparing against the RL-based refinement method MaskRegulate (Xue et al., 2024) to demonstrate effectiveness.

Table 1: Comparison of macro HPWL values ($\times 10^5$) of layouts generated by baseline placement methods and their corresponding refined results. Columns labeled "+ER" report the HPWL after applying EfficientRefiner. Values in parentheses represent the improvement rate achieved after refinement.

| Method | adaptec1 | adaptec1+ER | adaptec2 | adaptec2+ER | adaptec3 | adaptec3+ER | adaptec4 | adaptec4+ER |
|---|---|---|---|---|---|---|---|---|
| NTUPlace3 | 14.35 | 7.72 (-46.20%) | 65.33 | 41.04 (-37.18%) | 74.66 | 60.39 (-19.11%) | 63.21 | 48.46 (-23.33%) |
| DreamPlace 4.0 | 8.32 | 5.91 (-28.97%) | 38.68 | 30.27 (-27.14%) | 45.93 | 43.85 (-4.53%) | 39.78 | 35.83 (-9.93%) |
| DreamPlace 4.1.0 | 6.89 | 6.13 (-11.03%) | 50.09 | 35.11 (-21.92%) | 50.77 | 49.43 (-2.64%) | 40.92 | 38.01 (-7.11%) |
| WireMask-EA | 6.10 | 5.58 (-8.52%) | 54.78 | 51.93 (-5.20%) | 59.40 | 60.01 (+1.03%) | 59.46 | 53.51 (-10.01%) |
| MaskPlace | 6.69 | 5.98 (-10.67%) | 78.58 | 55.34 (-29.57%) | 118.18 | 89.43 (-24.33%) | 91.22 | 62.92 (-31.02%) |
| Chipformer | 7.13 | 5.94 (-16.69%) | 64.42 | 47.00 (-27.04%) | 80.55 | 63.32 (-21.39%) | 68.73 | 52.77 (-23.22%) |
| EfficientPlace | 6.14 | 5.47 (-10.91%) | 45.94 | 36.76 (-19.98%) | 57.37 | 54.40 (-5.18%) | 59.07 | 54.23 (-8.19%) |

| Method | bigblue1 | bigblue1+ER | bigblue2 | bigblue2+ER | bigblue3 | bigblue3+ER | bigblue4 | bigblue4+ER |
|---|---|---|---|---|---|---|---|---|
| NTUPlace3 | 6.74 | 4.08 (-39.47%) | 12.17 | 9.03 (-25.80%) | 60.78 | 32.37 (-46.74%) | 95.30 | 60.16 (-39.87%) |
| DreamPlace 4.0 | 2.36 | 2.13 (-9.75%) | 7.33 | 6.87 (-6.28%) | 239.72 | 217.96 (-9.08%) | 390.94 | 164.37 (-57.96%) |
| DreamPlace 4.1.0 | 2.41 | 2.21 (-8.30%) | 7.62 | 7.66 (+0.52%) | 25.32 | 25.77 (+1.78%) | 64.14 | 58.42 (-8.92%) |
| WireMask-EA | 2.17 | 2.22 (+2.30%) | 11.23 | 10.61 (-5.52%) | 67.17 | 39.20 (-41.64%) | 79.65 | 64.82 (-18.62%) |
| MaskPlace | 2.67 | 2.31 (-13.48%) | 17.49 | 12.41 (-29.05%) | 62.90 | 37.04 (-41.11%) | 112.87 | 70.07 (-37.92%) |
| Chipformer | 3.09 | 2.63 (-14.89%) | 13.30 | 11.86 (-10.83%) | 81.77 | 36.53 (-55.33%) | 105.62 | 64.74 (-38.70%) |
| EfficientPlace | 2.29 | 2.23 (-2.62%) | 12.85 | 10.42 (-18.91%) | 58.15 | 43.62 (-24.99%) | 84.18 | 64.44 (-23.45%) |

**Macro HPWL Optimization.** Table 1 reports HPWL results before and after applying EfficientRefiner. EfficientRefiner achieves average HPWL reductions of 7.20%–34.71% across all circuits for the baseline methods listed in the table. All refinements complete within 10 minutes as shown in Table 11, highlighting the efficiency of our approach. Moreover, EfficientRefiner scales to larger designs than RL-based methods due to its efficiency and scalability. We test on the ICCAD2015 benchmark with 8192 modules and thousands of fixed ports to validate its efficiency, with the results provided in Appendix A.4.1. The refinement time is shown in Table 12.

Table 2: Comparison of surrogate metrics and PPA results before and after refinement on ChiPBench circuits. "DP" denotes DreamPlace 4.1.0, and "DP+ER" denotes DreamPlace refined with EfficientRefiner. The best results are marked in **bold**.

| Circuit | Method | HPWL↓ | WL↓ | Cong↓ | Power↓ | NVP↓ | WNS↑ | TNS↑ | Area↓ |
|---|---|---|---|---|---|---|---|---|---|
| ariane136 | DP | 6211190 | **7370520** | **0.2481** | **0.3836** | 1842 | -0.2471 | -208.74 | 393322 |
| | DP+ER | **6133533** | 7430453 | 0.2502 | 0.3847 | **1779** | **-0.2277** | **-166.55** | **393161** |
| bp_fe | DP | 2246648 | 2817587 | 0.4943 | 0.1655 | 177 | -0.6845 | -40.16 | 71872 |
| | DP+ER | **2204814** | **2692443** | **0.4692** | **0.1652** | **112** | **-0.3469** | **-19.09** | **71596** |
| bp_be | DP | 3429613 | 4223729 | 0.5977 | 0.1466 | **111** | -0.6366 | -52.07 | 123881 |
| | DP+ER | **3230676** | **3886870** | **0.5972** | **0.1427** | **111** | **-0.6184** | **-49.00** | **121749** |
| bp_be12 | DP | 3659015 | 4187820 | 0.5108 | 0.0753 | 115 | -0.6826 | -65.89 | **92695** |
| | DP+ER | **3560677** | **4097495** | **0.4998** | **0.0752** | **114** | **-0.6015** | **-54.64** | 92827 |
| bp_multi57 | DP | 6668232 | 7485321 | 0.5235 | **0.1055** | 457 | -2.8632 | -799.80 | 210043 |
| | DP+ER | **5972371** | **6714072** | **0.4702** | 0.1059 | **411** | **-2.5053** | **-622.87** | **204627** |
| bp68 | DP | 12744064 | 14728606 | 0.4597 | 0.1530 | 2427 | -2.9514 | -1153.07 | 275709 |
| | DP+ER | **11186402** | **12856599** | **0.4037** | **0.1485** | **563** | **-2.1447** | **-746.56** | **269561** |
| swerv_wrapper | DP | 4642293 | 5481023 | 0.3918 | 0.2743 | 1421 | -0.6348 | -543.29 | 230130 |
| | DP+ER | **4351614** | **5139469** | **0.3532** | **0.2680** | **1296** | **-0.5787** | **-459.99** | **228604** |
| VeriGPU | DP | 1186895 | 1674544 | **0.1838** | 0.0951 | 1650 | -0.5759 | -210.83 | 153312 |
| | DP+ER | **1174880** | **1656132** | 0.1900 | **0.0900** | **531** | **-0.3665** | **-66.44** | **152468** |

**Mixed-Size HPWL and PPA Evaluation.** We evaluate mixed-size and PPA performance on PPA supported circuits from ChiPBench and ICCAD2015. Results on ChiPBench are shown in Table 2. As shown in the ChiPBench paper, most existing macro placement methods focus on macro HPWL optimization and provide limited improvements on PPA metrics. So we adopt the state-of-the-art mixed-size placer DreamPlace 4.1.0 as the baseline and refine its macro placements to provide a stronger comparison. Details of the refinement process and PPA evaluation process are provided in Appendix A.3. Experimental results show consistent improvements: mixed-size HPWL is reduced by 5% on average, while WNS and TNS improve by 20% and 29%, respectively. Additional results on ICCAD2015 presented in Appendix A.4.2 further confirm the effectiveness of our refinement.

**Comparison with RL-based Adjustment Method.** We compare EfficientRefiner with the RL-based method MaskRegulate (Xue et al., 2024), both applied to refine ICCAD2015 layouts generated

by DreamPlace 4.0. As shown in Table 3, EfficientRefiner consistently delivers higher placement quality, reducing mixed-size HPWL by 27% on average. While MaskRegulate requires 30+ hours of training for 1k iterations, our method completes 5k refinement iterations in only about 3 minutes.

Table 3: Comparison of EfficientRefiner (ER) with MaskRegulate (MR) on the ICCAD2015 benchmark. "DP" denotes DreamPlace 4.0. HPWL values are reported in units of $10^8$. The best results are marked in **bold**.

| Circuit | Method | HPWL | WNS* | TNS* | Circuit | Method | HPWL | WNS* | TNS* |
|---|---|---|---|---|---|---|---|---|---|
| superblue1 | DP | 12.91 | -3583.26 | -827.03 | superblue7 | DP | 13.74 | -2082.79 | -152.93 |
| | DP+MR | 6.21 | -1241.74 | -51.75 | | DP+MR | 8.20 | -1852.93 | -58.65 |
| | DP+ER | **4.46** | **-210.98** | **-22.80** | | DP+ER | **6.58** | **-304.41** | **-20.20** |
| superblue3 | DP | 11.15 | -785.77 | -93.10 | superblue10 | DP | 13.74 | -2082.79 | -152.93 |
| | DP+MR | 7.42 | -886.39 | **-88.56** | | DP+MR | 12.16 | -3215.87 | -142.13 |
| | DP+ER | **5.17** | **-158.40** | -114.95 | | DP+ER | **7.56** | **-696.24** | **-22.56** |
| superblue4 | DP | 7.70 | -1211.13 | -49.75 | superblue16 | DP | 11.47 | -4039.89 | -253.56 |
| | DP+MR | 4.24 | -912.54 | -45.85 | | DP+MR | **4.32** | **-522.66** | **-41.76** |
| | DP+ER | **3.46** | **-319.83** | **-25.75** | | DP+ER | 5.44 | -1409.37 | -36.81 |
| superblue5 | DP | 10.33 | -1009.39 | -70.33 | superblue18 | DP | 4.42 | -181.41 | -80.20 |
| | DP+MR | 7.34 | -667.19 | -77.50 | | DP+MR | 3.10 | -415.80 | -29.67 |
| | DP+ER | **4.67** | **-258.66** | **-54.29** | | DP+ER | **2.33** | **-128.55** | **-18.47** |

Note: As the ICCAD2015 benchmark is not supported by OpenRoad, the WNS and TNS values are estimated pre-routing using OpenTimer.

## 5.3 ANALYSIS

**Effectiveness of GPU Programming.** We compare our Numba-based GPU implementation with a PyTorch-based version to demonstrate the benefits of parallel computation. As shown in Table 13, our approach achieves over $1000\times$ speedup. This improvement stems from the fact that, when maintaining the same space complexity in overlap computation, the PyTorch implementation can only process overlaps sequentially for each module pair. In contrast, our implementation computes overlaps for multiple module pairs in parallel to yield significant performance gains.

**Effectiveness of the Pruning Strategy.** To evaluate the efficiency improvement of the pruning strategy, we conduct experiments on the ICCAD2015 dataset refining 8192 modules along with several thousand ports. The runtime comparison with or without the pruning strategy is presented in Table 12, showing that this strategy achieves an average speedup of $8\times$.

**Effectiveness of Fine-grained Overlap Modeling.** We compare EfficientRefiner with the analytical approaches NTUPlace3 and DreamPlace to evaluate the effectiveness of our fine-grained module-pair overlap formulation and to examine the impact of legalization, which often degrades solution quality. In this experiment, all methods are applied to refine macro layouts generated by EfficientPlace. Both baselines rely on coarse-grained density formulations for overlap removal. We use the HPWL metric as an indicator to measure the effect of legalization. Table 9 reports the overlap rates before legalization, and Table 10 shows the relative HPWL increase after legalization. Across all eight benchmarks, EfficientRefiner achieves near-zero overlap, significantly outperforming NTUPlace3 and DreamPlace. Moreover, it yields the lowest average HPWL increase of only 0.81%, compared to 8.60% for NTUPlace3 and 57.67% for DreamPlace.

**Parameter Analysis.** We investigate the impact of the overlap weight $\alpha$ in Eq. (2) on the refinement process, using HPWL as the optimization objective for demonstration. Figures 9–12 in Appendix A.4.6 show HPWL changes before and after legalization (left) and overlap rates before legalization (right) for various $\alpha$ values. Similar trends are observed across different baselines and circuits: (1) very small $\alpha$ (e.g., $< 10$) may cause legalization failure. (2) Moderate $\alpha$ (10–100) leads to high overlap and a large HPWL increase after legalization. (3) Larger $\alpha$ ($> 100$) reduces overlap and lowers the legalization impact on HPWL. (4) The HPWL value remains small for $\alpha > 5k$.

## 6 CONCLUSION

This paper presented EfficientRefiner, an analytical-based framework for refining macro placements produced by existing placement methods. The method leverages the strengths of analytical techniques while being tailored to the macro refinement setting. It adjusts macro positions using a comprehensive representation of the full layout and incorporates a fine-grained pairwise overlap ob-

jective that effectively reduces module overlaps without inducing excessive spreading. Moreover, it accelerates refinement with pruning strategy and GPU-based parallel computation, substantially improving efficiency. Experimental results show that EfficientRefiner achieves notable improvements in both HPWL and PPA over existing methods.

For future work, we aim to integrate more accurate PPA-related metrics into the optimization objective. For example, LaMPlace (Geng et al., 2025) introduces learned PPA predictors that could be incorporated into our framework. However, since the data released in the LaMPlace GitHub repository is currently incomplete, we leave this direction to future work. We also plan to conduct more rigorous PPA evaluations on the large-scale ICCAD2015 benchmark once commercial design tools become available.

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

# A APPENDIX

## A.1 MACRO PLACEMENT METRICS

We present the PPA metrics for chip placement below and discuss two key surrogate metrics, HPWL and regularity.

**PPA.** PPA (performance, power, and area) metrics are comprehensive indicators of the chip design quality. PPA consists of timing and physical metrics. In particular, timing performance is commonly assessed using worst negative slack (WNS) and total negative slack (TNS). Slack represents the difference between the required arrival time of a signal at a circuit endpoint and its actual arrival time. A negative slack indicates that the timing constraint is violated. WNS captures the most critical violation in the design by reporting the worst slack value, whereas TNS measures the overall severity of timing issues by summing all negative slack values. Together with power consumption and area utilization, these metrics provide a practical basis for assessing design quality.

**HPWL.** HPWL is a widely adopted metric for efficiently approximating total wirelength. A lower HPWL often indicates reduced routing resource usage and improved performance. The HPWL of the placement is computed as the sum of the half-perimeters of all net bounding boxes, i.e., the smallest rectangles enclosing all pins in each net, as shown in Eq.(10).

$$\text{HPWL} = \sum_{e \in E} (\max_{p \in e} p_x - \min_{p \in e} p_x + \max_{p \in e} p_y - \min_{p \in e} p_y) \tag{10}$$

In the above equation, $p$ represents a pin belonging to net $e$. Its position, denoted as $(p_x, p_y)$, is determined by the position of its associated module $M_i$ plus the pin's offset $(\Delta p_x, \Delta p_x)$, as shown in Eq.(11).

$$(p_x, p_y) = (x_i, y_i) + (\Delta p_x, \Delta p_x), \quad p \in M_i \tag{11}$$

Since HPWL is not differentiable, the weighted-average function (Hsu et al., 2013) is commonly employed as an approximation to facilitate gradient descent optimization. The approximated HPWL in the x-direction is computed as shown in Eq.(12), with the estimation in the y-direction derived similarly. In this equation, for each net $e \in E$, the minuend and subtrahend estimate the upper and lower boundaries of $e$'s bounding box, respectively. $\gamma$ is a hyperparameter that governs the trade-off between accuracy and smoothness in HPWL estimation. A smaller $\gamma$ yields a more precise approximation but reduces the smoothness of the function. Eq.(12) is also utilized in our method for HPWL estimation.

$$\hat{\text{HPWL}}_x = \sum_{e \in E} \left( \frac{\sum_{p \in e} p_x e^{\frac{p_x}{\gamma}}}{\sum_{p \in e} e^{\frac{p_x}{\gamma}}} - \frac{\sum_{p \in e} p_x e^{-\frac{p_x}{\gamma}}}{\sum_{p \in e} e^{-\frac{p_x}{\gamma}}} \right) \tag{12}$$

**Regularity.**   Regularity encourages macros to be placed closer to the chip boundary, thereby leaving larger spaces available for standard cells placement. Incooperating this metric as an optimization objective has been shown in prior work (Xue et al., 2024) to be beneficial for both mixed-size placement and overall PPA. The regularity of a macro located at position $(x, y)$ is defined as $min\{x, R_w - x\} + min\{y, R_h - y\}$, where $R_w$ and $R_h$ denote the width and height of the chip, respectively.

## A.2   ALGORITHMS FOR GPU PROGRAMMING ACCELERATION

The pseudocode for computing the overlapping lengths and their gradients for each module pair is presented in Algorithms 1 and 2. In Algorithm 1, the pairwise overlap gradient between $(M_i, M_j)$ is computed as Eq.(13).

$$\frac{\partial \text{Over}\hat{\text{l}}\text{ap}_{ijx}}{\partial x_i} = \begin{cases} 1, & x_i < x_j \quad and \quad |x_i - x_j| < w_i + w_j \\ -1, & x_i \geq x_j \quad and \quad |x_i - x_j| < w_i + w_j \\ 0, & |x_i - x_j| \geq w_i + w_j \end{cases} \tag{13}$$

Algorithm 3 outlines the computation of the overall objective function and its derivatives with respect to module positions.

---

**Algorithm 1** GetOverlap

---

**Input:** Module pair $(M_i, M_j)$ with sizes $(w_i, h_i), (w_j, h_j)$, and positions $(x_i, y_i), (x_j, y_j)$, respectively.

**Output:** The overlapping length between $M_i$ and $M_j$ along $x$- and $y$- directions, denoted as $\text{Over}\hat{\text{l}}\text{ap}_{xij}$ and $\text{Over}\hat{\text{l}}\text{ap}_{yij}$, respectively.

1:   $\delta\text{Over}\hat{\text{l}}\text{ap}_{xij} \leftarrow 0$
2: **if** $i \neq j$ and $-\frac{w_i+w_j}{2} < x_i - x_j < \frac{w_i+w_j}{2}$ **then**
3:     $\{M_i$ overlap with $M_j$ in the $x$-direction$\}$
4:     **if** $x_i < x_j$ **then**
5:        $\text{Over}\hat{\text{l}}\text{ap}_{xij} \leftarrow (x_i + \frac{w_i}{2}) - (x_j + \frac{w_j}{2})$
6:     **else**
7:        $\text{Over}\hat{\text{l}}\text{ap}_{\text{x}ij} \leftarrow (x_j + \frac{w_j}{2}) - (x_i + \frac{w_i}{2})$
8:     **end if**
9: **end if**
10:
11: $\delta\text{Over}\hat{\text{l}}\text{ap}_{yij} \leftarrow 0$
12: **if** $i \neq j$ and $-\frac{h_i+h_j}{2} < y_i - y_j < \frac{h_i+h_j}{2}$ **then**
13:     $\{M_i$ overlap with $M_j$ in the $y$-direction$\}$
14:     **if** $y_i < y_j$ **then**
15:        $\text{Over}\hat{\text{l}}\text{ap}_{yij} \leftarrow (y_i + \frac{h_i}{2}) - (y_j + \frac{h_j}{2})$
16:     **else**
17:        $\text{Over}\hat{\text{l}}\text{ap}_{yij} \leftarrow (y_j + \frac{h_j}{2}) - (y_i + \frac{h_i}{2})$
18:     **end if**
19: **end if**

**Return:** $\text{Over}\hat{\text{l}}\text{ap}_{xij},$    $\text{Over}\hat{\text{l}}\text{ap}_{yij}$

---

## A.3   EXPERIMENTAL SETTINGS

**Code.**   The code is provided at https://anonymous.4open.science/r/EfficientRefiner-100D.

**Benchmark Statistics.**   The statistics of the ISPD2005, ICCAD2015, and ChiPBench circuits are reported in Tables 4, 5, and 6. The column "Macros (to place)" denotes the number of macros considered for placement. For the ICCAD2015 benchmarks, we additionally perform refinement on 8192 modules, a scale considerably larger than that handled by existing RL-based macro placement methods.

**Parameter Settings.**   We set the number of refinement iterations to 50k on the ISPD2005 benchmark to achieve better HPWL results. However, we found 5k iterations are already sufficient for

**Algorithm 2** GetOverlapGrad

**Input:** Module pair $(M_i, M_j)$ with sizes $(w_i, h_i), (w_j, h_j)$, and positions $(x_i, y_i), (x_j, y_j)$, respectively.

**Output:** The gradient of the overlapping length between $M_i$ and $M_j$ with respect to $x_i$ along $x$- and $y$- directions, denoted as $\delta\hat{\text{Overlap}}_{xij}$ and $\delta\hat{\text{Overlap}}_{yij}$, respectively.

1: $\delta\hat{\text{Overlap}}_{xij} \leftarrow 0$
2: **if** $i \neq j$ and $-\frac{w_i+w_j}{2} < x_i - x_j < \frac{w_i+w_j}{2}$ **then**
3:    $\{M_i$ overlap with $M_j$ in the $x$-direction$\}$
4:    **if** $x_i < x_j$ **then**
5:       $\delta\hat{\text{Overlap}}_{xij} \leftarrow -1$
6:    **else**
7:       $\delta\hat{\text{Overlap}}_{xij} \leftarrow 1$
8:    **end if**
9: **end if**
10:
11: $\delta\hat{\text{Overlap}}_{yij} \leftarrow 0$
12: **if** $i \neq j$ and $-\frac{h_i+h_j}{2} < y_i - y_j < \frac{h_i+h_j}{2}$ **then**
13:    $\{M_i$ overlap with $M_j$ in the $y$-direction$\}$
14:    **if** $y_i < y_j$ **then**
15:       $\delta\hat{\text{Overlap}}_{yij} \leftarrow -1$
16:    **else**
17:       $\delta\hat{\text{Overlap}}_{yij} \leftarrow 1$
18:    **end if**
19: **end if**
**Return:** $\delta\hat{\text{Overlap}}_{xij}$,   $\delta\hat{\text{Overlap}}_{yij}$

Table 4: Statistics of the ISPD2005 Circuit Benchmark

| Circuit | Macros | Macros(to place) | Macro-related Nets | Standard Cells | Nets | Area Util%) |
|---|---|---|---|---|---|---|
| adaptec1 | 543 | 543 | 693 | 210904 | 221142 | 55.62 |
| adaptec2 | 566 | 566 | 4201 | 254457 | 266009 | 74.46 |
| adaptec3 | 723 | 723 | 3259 | 450927 | 466758 | 61.51 |
| adaptec4 | 1329 | 1329 | 2949 | 494716 | 515951 | 48.62 |
| bigblue1 | 560 | 560 | 409 | 277604 | 284479 | 31.58 |
| bigblue2 | 23084 | 1024 | 33223 | 534782 | 577235 | 32.43 |
| bigblue3 | 1293 | 1293 | 3937 | 1095519 | 1123170 | 66.81 |
| bigblue4 | 8170 | 1024 | 22223 | 2169183 | 2229886 | 35.68 |

Table 5: Statistics of the ICCAD2015 Circuit Benchmark

| Circuit | Macros (to Place) | Standard Cells | Nets | Pins | Ports | Area Util(%) |
|---|---|---|---|---|---|---|
| superblue1 | 512 | 1215820 | 1215710 | 3767494 | 6528 | 85 |
| superblue3 | 512 | 1219170 | 1224979 | 3905321 | 6482 | 87 |
| superblue4 | 512 | 801968 | 802513 | 2497940 | 6623 | 90 |
| superblue5 | 512 | 1090247 | 1100825 | 3246878 | 4129 | 85 |
| superblue7 | 512 | 1937699 | 1933945 | 6372094 | 6501 | 90 |
| superblue10 | 512 | 984379 | 1898119 | 5560506 | 12257 | 87 |
| superblue16 | 512 | 985909 | 999902 | 3013268 | 4449 | 85 |
| superblue18 | 512 | 771845 | 771542 | 2559143 | 3978 | 85 |

Table 6: Statistics of Circuits in ChiPBench

| Design | Macros | Standard Cells | Nets | Pins | Ports |
|---|---|---|---|---|---|
| ariane136 | 136 | 171347 | 201428 | 1000876 | 495 |
| bp_fe | 11 | 33188 | 39512 | 185524 | 2511 |
| bp_be | 10 | 51382 | 62228 | 293276 | 3029 |
| swerv_wrapper | 28 | 98039 | 113582 | 573688 | 1416 |
| dft68 | 68 | 41974 | 56217 | 226420 | 132 |
| bp68 | 68 | 164039 | 191475 | 887046 | 1198 |
| VeriGPU | 12 | 71082 | 85081 | 421857 | 134 |
| bp_be12 | 12 | 38393 | 47030 | 220938 | 3029 |

---

**Algorithm 3** Calculation of the Overlap Objective Function and Its Gradient

---

**Input:** The set of modules $\{M_1, M_2, ..., M_n\}$, with module sizes $\{(w_1, h_1), (w_2, h_2), ..., (w_n, h_n)\}$, and module positions $\{(x_1, y_1), (x_2, y_2), ..., (x_n, y_n)\}$.

**Output:** The overlap function $\hat{\text{Overlap}}$, and the derivative of the overlap function with respect to the module positions $\delta\hat{\text{Overlap}} = (\frac{\delta\hat{\text{Overlap}}}{\delta x_1}, \frac{\delta\hat{\text{Overlap}}}{\delta y_1}, \frac{\delta\hat{\text{Overlap}}}{\delta x_2}, \frac{\delta\hat{\text{Overlap}}}{\delta y_2}, ..., \frac{\delta\hat{\text{Overlap}}}{\delta x_n}, \frac{\delta\hat{\text{Overlap}}}{\delta y_n})$

1: Divide the placement region into $B \times B$ bins, each bin with size $(b_w, b_h)$.
2: {Categorize modules into big modules and small modules according to the bin size}
3: bigModules $\leftarrow \{\}$, smallModules $\leftarrow \{\}$
4: **for** each thread $0 \le t < n$ **do**
5:   **if** $w_t \le b_w$ and $h_t \le b_h$ **then**
6:     smallModules $\leftarrow$ smallModules $\cup \{M_t\}$
7:   **else**
8:     bigModules $\leftarrow$ bigModules $\cup \{M_t\}$
9:   **end if**
10: **end for**
11:
12: {Retrieve the modules contained in each bin}
13:   $bins[i] \leftarrow \{\}, \forall i \in [0, B \times B - 1]$
14: **for** each thread $0 \le t < n$ **do**
15:   **if** $M_t \in$ smallModules **then**
16:     $x_b \leftarrow \lfloor x_t/b_w \rfloor, y_b \leftarrow \lfloor y_t/b_h \rfloor$
17:     $i \leftarrow x_b \cdot B + y_b$
18:     $bins[i] \leftarrow bins[i] \cup \{M_t\}$
19:   **end if**
20: **end for**
21:
22: {Calculate overlap and its gradient in the $x$-direction}
23: $\hat{\text{Overlap}} \leftarrow 0, \quad \delta\hat{\text{Overlap}}[i] \leftarrow 0, \forall i \in [0, 2n]$
24: **for** each thread $0 \le t < n^2$ **do**
25:   $i \leftarrow t/n, j \leftarrow t \mod n$
26:   **if** $M_i \in$ bigModules **then**
27:     {Calculate overlapping length and its gradient between $M_i$ and all other modules}
28:     **if** $M_j \in$ bigModules and $i < j$ or $M_j \in$ smallModules **then**
29:       {Compute overlapping length with Algorithm 1}
30:       $\hat{\text{Overlap}}_{xij}, \hat{\text{Overlap}}_{yij} \leftarrow GetOverlap(x_i, x_j, w_i, w_j)$
31:       {Compute gradient with Algorithm 2}
32:       $\delta\hat{\text{Overlap}}_{xij}, \delta\hat{\text{Overlap}}_{yij} \leftarrow GetOverlapGrad(x_i, x_j, w_i, w_j)$
33:       $\hat{\text{Overlap}} \overset{at.}{\leftarrow} \hat{\text{Overlap}} + \hat{\text{Overlap}}_{xij} * \hat{\text{Overlap}}_{yij}$         {Atomic add}
34:       $\delta\hat{\text{Overlap}}[2i] \overset{at.}{\leftarrow} \hat{\text{Overlap}}[2i] + \delta\hat{\text{Overlap}}_{xij} * \hat{\text{Overlap}}_{yij}$
35:       $\delta\hat{\text{Overlap}}[2i + 1] \overset{at.}{\leftarrow} \hat{\text{Overlap}}[2i + 1] + \delta\hat{\text{Overlap}}_{yij} * \hat{\text{Overlap}}_{xij}$
36:     **end if**
37:   **else**
38:     Calculate overlapping length and its gradient between $M_i$ and the modules within modules located in the same bin or any of its eight adjacent neighboring bins
39:     $x_b \leftarrow \lfloor x_t/b_w \rfloor, y_b \leftarrow \lfloor y_t/b_h \rfloor$         {Locate the bin containing $M_i$}
40:     **for** $offset_x, offset_x$ in $\{-1, 0, 1\}$ **do**
41:       $b \leftarrow (x_b + offset_x) \cdot B + (y_b + offset_y)$       {Get Neighboring bin}
42:       **if** $M_j \in bins[b]$ and $i < j$ **then**
43:         Compute overlap and its gradient as line 29-35.
44:       **end if**
45:     **end for**
46:   **end if**
47: **end for**

**Return:** $\hat{\text{Overlap}}, \quad \delta\hat{\text{Overlap}}$

---

strong performance so we use 5k iterations on both the ICCAD2015 and ChiPBench datasets. The weight for overlap in Eq.(2) is set to $10^5$ in all experiments (except for parameter analysis), while the weight for regularity is set to 2.

**Experimental Platform.** Refinements for 512 macros are conducted on a standardized platform equipped with an NVIDIA GeForce RTX 2080 Ti GPU. Refinement for 8192 macros and training for other baselines are executed on a server equipped with a NVIDIA RTX 3090Ti GPU and 40 Intel Xeon Silver 4210R CPUs (2.40 GHz).

**Settings for Baseline Methods.** The baseline methods for refinement include the RL-based methods MaskPlace Lai et al. (2022), Chipformer Lai et al. (2023), and EfficientPlace Geng et al. (2024); the stochastic-based method WireMask-EA Shi et al. (2023); and the analytical-based methods DreamPlace Lin et al. (2019) and NTUPlace3 Chen et al. (2008). We also compare our EfficientRefiner with RL-based refinement method MaskRegulate (Xue et al., 2024). The specific settings for running each baseline method are as follows:

- DreamPlace: We run the released code of DreamPlace 4.0 and 4.1.0 with the default parameters.

- NTUPlace3: We use the released binary file of NTUPlace3 for execution.

- WireMask-EA: We run the released code of WireMask-EA with default parameters, iterating for 1000 rounds.

- EfficientPlace: We run the released code of EfficientPlace with default parameters, iterating for 1000 rounds.

- MaskPlace: We run the released code of MaskPlace with default parameters, iterating for 3000 rounds.

- Chipformer: We execute the released code of Chipformer. We use the pretrained model parameters provided in the GitHub repository, fine-tune the Online Decision Transformer for 300 rounds with the default configuration.

- MaskRegulate: We use the released implementation of MaskRegulate with the pretrained model parameters provided in its GitHub repository. Since the released code is not directly compatible with DreamPlace 4.1.0, the initial placement for adjustment is generated using DreamPlace 4.0.

**Procedure for Refinement on Mixed-Size Layouts and PPA Evaluation.** We first extract the macros for refinement from mixed-size layouts generated by existing methods. Then we fix refined macros and place the standard cells with DreamPlace.

The process for PPA evaluation is described as follows. For ChiPBench circuits, we feed layouts into the ChiPBench flow, which uses the OpenROAD (Ajayi & Blaauw, 2019) tool chain for detailed placement, routing, and metric evaluation. For the ICCAD2015 benchmark, since it does not support the required technology files for the open source OpenROAD tool and the commercial PPA evaluation tools are not accessible for us, we employ OpenTimer (Huang & Wong, 2015) to estimate PPA in the same way as DreamPlace. As this estimation does not include routing, the results are for reference.

## A.4 Additional Results

### A.4.1 Macro Results on ICCAD2015 Benchmark

HPWL results for refining circuits in the ICCAD2015 benchmarks is reported in Table 7, the results show that our method achieves 18% decrease in HPWL.

Table 7: HPWL comparison ($\times 10^5$) between DreamPlace 4.1.0 (DP) and DP with EfficientRefiner (DP+ER) on the ICCAD2015 benchmark.

| Circuits | superblue1 | superblue3 | superblue4 | superblue5 | superblue7 | superblue10 | superblue16 | superblue18 |
|----------|-----------|-----------|-----------|-----------|-----------|-------------|-------------|-------------|
| DP | 8.29 | 19.34 | 44.02 | 43.41 | 35.51 | 48.38 | **16.39** | **13.47** |
| DP+ER | **7.43** | **15.70** | **33.61** | **36.75** | **25.73** | **33.42** | 16.62 | 10.81 |

### A.4.2 MIXED-SIZE RESULTS ON THE ICCAD2015 BENCHMARK

The mixed-size placement results on ICCAD2015 benchmarks are shown in Table 8. We refine the layouts generated by DreamPlace 4.1.0 by first extracting the macros, refining them, and then placing the remaining standard cells with DreamPlace. Our method improves the placement quality on 7 out of 8 circuits, achieving an average 34% reduction in mixed-size HPWL. It should be noted that, since the open-source OpenROAD tool does not support ICCAD2015 benchmarks and commercial software is currently unavailable to us, the reported PPA results are estimated by OpenTimer based on the placement. As these estimates are obtained without post-routing, they are not exact and should be regarded as references. For accurate results, we refer to the evaluation on ChiPBench circuits in the main text.

Table 8: Mixed-size placement results on the ICCAD2015 benchmark. "DP" denotes DreamPlace 4.1.0, and "DP+ER" denotes DreamPlace refined with EfficientRefiner. The best results are marked in **bold**.

| Circuit | Method | HPWL ($\times 10^8$) | WNS* | TNS* |
|---|---|---|---|---|
| superblue1 | DP | 8.33 | -2048.83 | -57.04 |
| | DP+ER | **4.31** | **-275.92** | **-19.26** |
| superblue3 | DP | 8.97 | -1062.84 | -92.06 |
| | DP+ER | **4.79** | **-125.89** | **-29.13** |
| superblue4 | DP | 3.43 | -289.19 | -18.33 |
| | DP+ER | **3.14** | **-231.28** | **-17.85** |
| superblue5 | DP | 7.07 | -426.30 | -58.88 |
| | DP+ER | **4.54** | **-95.89** | **-46.24** |
| superblue7 | DP | 14.19 | -779.96 | -36.19 |
| | DP+ER | **6.01** | **-210.53** | **-16.98** |
| superblue10 | DP | 10.78 | -957.88 | -49.88 |
| | DP+ER | **7.58** | **-512.87** | **-19.45** |
| superblue16 | DP | 6.43 | -829.18 | -41.20 |
| | DP+ER | **3.97** | **-377.07** | **-18.73** |
| superblue18 | DP | **2.38** | **-48.39** | **-12.29** |
| | DP+ER | 2.43 | -170.00 | -12.78 |

Note: As the ICCAD2015 benchmark is not supported by the open-source tool OpenRoad, the WNS and TNS values are estimated pre-routing using OpenTimer.

### A.4.3 OPTIMIZATION OBJECTIVE TRENDS DURING REFINEMENT

Figure 5 shows the trend of the differentiable HPWL estimate ($\hat{\text{HPWL}}$) when HPWL is used as the optimization objective on superblue1 and superblue4. It can be seen that HPWL generally decreases as the number of refinement iterations increases.

Figures 6–8 show the trajectories of $\hat{\text{HPWL}}$ and $\text{Regularity}$ when optimizing both objectives jointly on superblue1 and superblue4. The effectiveness of our approach arises from both reducing macro $\hat{\text{HPWL}}$ and lowering $\text{Regularity}$ to provide sufficient placement space for standard cells.

### A.4.4 EFFECTIVENESS OF FINE-GRAINED OVERLAP MODELING

Overlap rate before legalization and HPWL increase rate after legalization for EfficientRefiner and analytical based methods DreamPlace and NTUPlace3 are shown in Table 9 and 10, respectively.

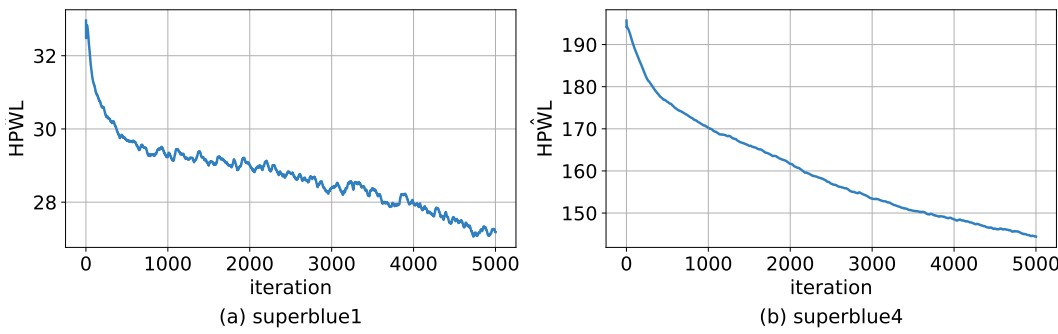

(a) superblue1          (b) superblue4

Figure 5: $\mathrm{HP\hat{W}L}$ **trend during refinement for (a) superblue1 and (b) superblue4**

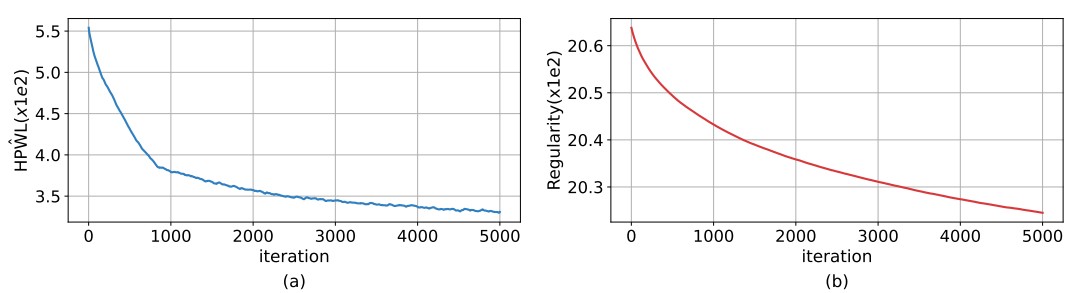

(a)          (b)

Figure 6: **Trends of optimization objectives during refinement on superblue1** (a) $\mathrm{HP\hat{W}L}$ trend. (b) Regularity trend.

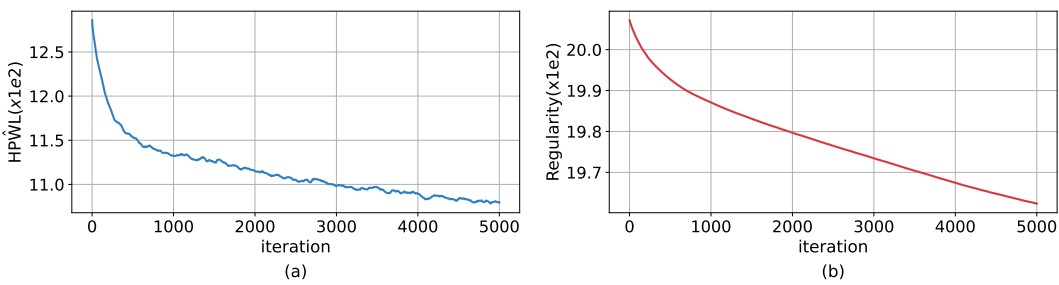

(a)          (b)

Figure 7: **Trends of optimization objectives during refinement on superblue3** (a) $\mathrm{HP\hat{W}L}$ trend. (b) Regularity trend.

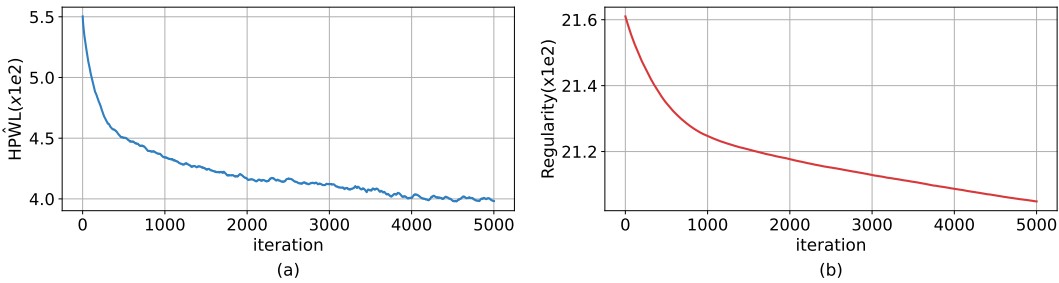

(a)          (b)

Figure 8: **Trends of optimization objectives during refinement on superblue4** (a) $\mathrm{HP\hat{W}L}$ trend. (b) Regularity trend.

Table 9: Overlap rate (%) before legalization for various methods on the ISPD 2005 dataset. The lowest overlap rate for each benchmark is highlighted in **bold**.

| Method | adaptec1 | adaptec2 | adaptec3 | adaptec4 | bigblue1 | bigblue2 | bigblue3 | bigblue4 |
|---|---|---|---|---|---|---|---|---|
| NTUPlace3 | 8.285 | 12.883 | 12.920 | 15.864 | 6.779 | 5.915 | 7.654 | 5.117 |
| DreamPlace | 5.300 | 4.228 | 4.362 | 4.362 | 8.794 | 5.818 | 18.789 | 15.120 |
| EfficientRefiner | **0.002** | **0.002** | **0.001** | **0.003** | **0.000** | **0.001** | **0.086** | **0.002** |

Table 10: HPWL increase rate (%) after legalization for various methods on the ISPD 2005 dataset. The lowest increase rate for each benchmark is highlighted in **bold**.

| Method | adaptec1 | adaptec2 | adaptec3 | adaptec4 | bigblue1 | bigblue2 | bigblue3 | bigblue4 |
|---|---|---|---|---|---|---|---|---|
| NTUPlace3 | 0.70 | 30.74 | 20.21 | 18.50 | 11.77 | 1.08 | **-25.25** | 11.07 |
| DreamPlace 4.0 | 62.50 | 54.10 | 6.05 | 4.93 | 22.92 | 13.64 | 261.84 | 35.38 |
| EfficientRefiner | **0.00** | **-0.05** | **0.15** | 0.18 | **-0.45** | **0.58** | 1.35 | **4.70** |

### A.4.5  ANALYSIS OF RUNTIME

Table 11 reports the runtime of EfficientPlace for 50k refinement iterations on the ISPD2005 dataset. Table 13 reports our runtime improvement over PyTorch implementation. Table 12 compares the refinement time with and without the acceleration technique.

Table 11: Time (in seconds) for refining 50k iterations on the ISPD 2005 dataset.

| adaptec1 | adaptec2 | adaptec3 | adaptec4 | bigblue1 | bigblue2 | bigblue3 | bigblue4 |
|---|---|---|---|---|---|---|---|
| 417 | 437 | 459 | 593 | 415 | 501 | 527 | 491 |

Table 12: Runtime comparison (in seconds) of EfficientRefiner (ER) with and without acceleration over 5k refinement iterations on the ICCAD2015 benchmark circuits.

| Circuit | superblue1 | superblue3 | superblue4 | superblue5 | superblue7 | superblue10 | superblue16 | superblue18 |
|---|---|---|---|---|---|---|---|---|
| ER w/o acceleration | 1125 | 1138 | 1173 | 922 | 1150 | 1775 | 955 | 901 |
| ER with acceleration | **135** | **135** | **153** | **136** | **138** | **174** | **142** | **130** |

Table 13: Runtime (s) comparison between our implementation and the PyTorch implementation, reporting the average runtime per iteration (over 10 iterations).

| Method | adaptec1 | adaptec2 | adaptec3 | adaptec4 |
|---|---|---|---|---|
| Our Implementation | **0.21** | **0.22** | **0.22** | **0.22** |
| PyTorch Implementation | 265.80 | 289.71 | 487.07 | 1663.23 |

### A.4.6  THE IMPACT OF OVERLAP RATE ON THE REFINEMENT PROCESS

Fig.9-12 show the HPWL growth before and after legalization, along with the overlap rate prior to legalization, for different values of overlap rate during refinement on the "adaptec1" and "adaptec3" layouts generated by EfficientPlace and DreamPlace.

### A.4.7  LAYOUT VISUALIZATIONS

Figure 13 and 14 provide visualizations of layouts before and after refinement for circuits in the IC-CAD2015 benchmark and ChiPBench, respectively. Figure 14 demonstrates reductions in both regularity and macro HPWL on ChipBench. In Fig.13, for the ICCAD2015 benchmarks "superblue1" to "superblue4", our refinement method preserves the general placement patterns produced by Dream-Place while improving both regularity and macro HPWL. The yellow boxes in the figures highlight that our method moves macros closer to the chip boundary, reducing the regularity metric and leaving more whitespace for standard-cell placement. The green boxes show that we also pull tightly connected macros closer together, thereby reducing macro HPWL.

In Fig.13, although MaskRegulate places many macros near the chip boundary, its regularity metric remains relatively high. A possible reason is that it often positions large macros along the outer boundary, which increases the average distance of small modules from the chip edge. Nevertheless, by pushing large macros outward and creating more whitespace for small modules, MaskRegulate still achieves better mixed-size results than DreamPlace.

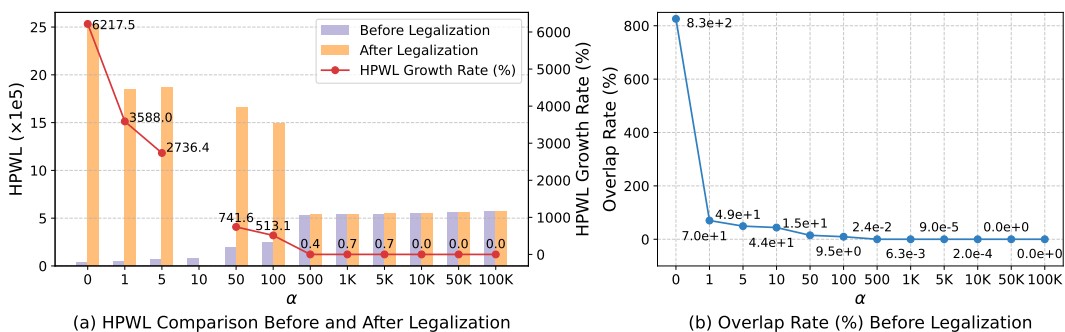

Figure 9: **The Impact of Parameter $\alpha$ on the Refinement Process for the "adaptec1" Layout Generated by EfficientPlace.** (a) Changes in HPWL before and after legalization for different values of $\alpha$. (b) Overlap rate before legalization for different values of $\alpha$.

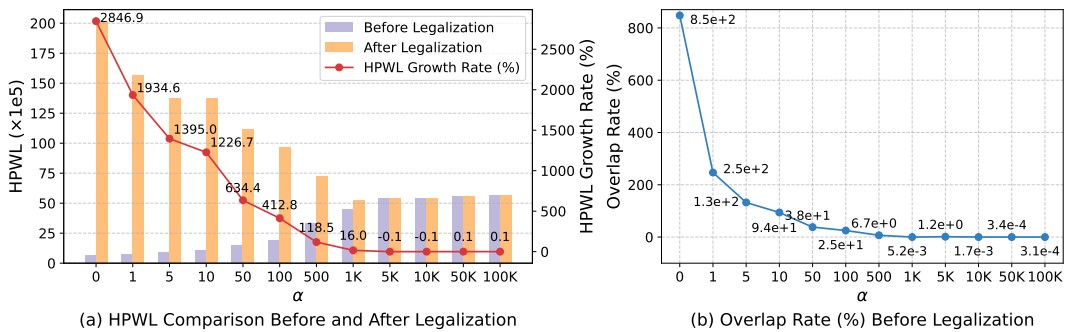

Figure 10: **The Impact of Parameter $\alpha$ on the Refinement Process for the "adaptec3" Layout Generated by EfficientPlace.**

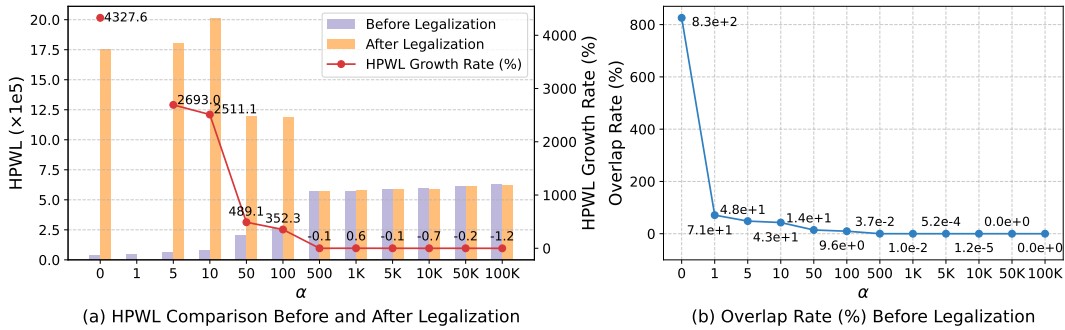

Figure 11: **The Impact of Parameter $\alpha$ on the Refinement Process for the "adaptec1" Layout Generated by DreamPlace 4.0.** Legalization failed for $\alpha = 1$.

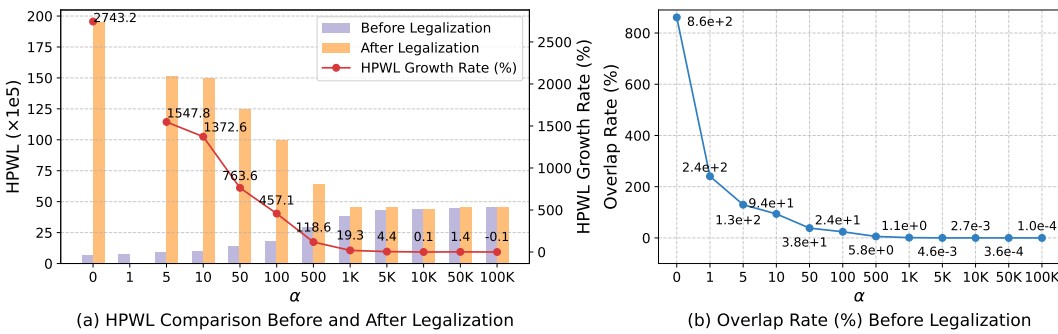

Figure 12: **The Impact of Parameter $\alpha$ on the Refinement Process for the "adaptec3" Layout Generated by DreamPlace 4.0.** Legalization failed for $\alpha = 1$.

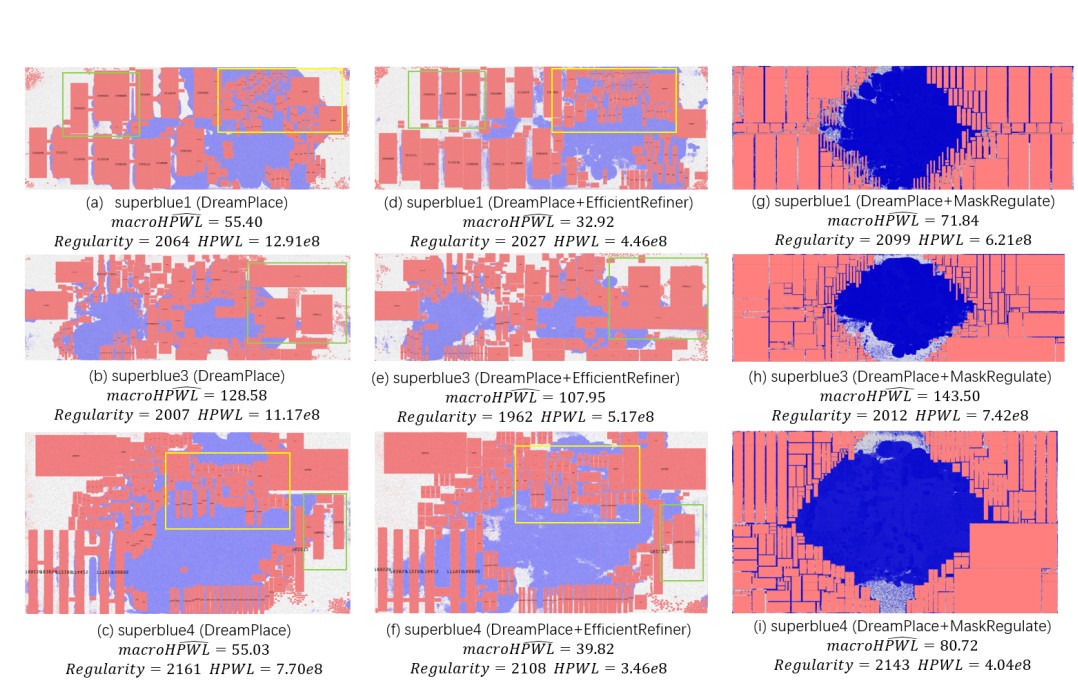

Figure 13: **Layout visualizations before and after refinement for the "superblue1"-"superblue3" circuits.**

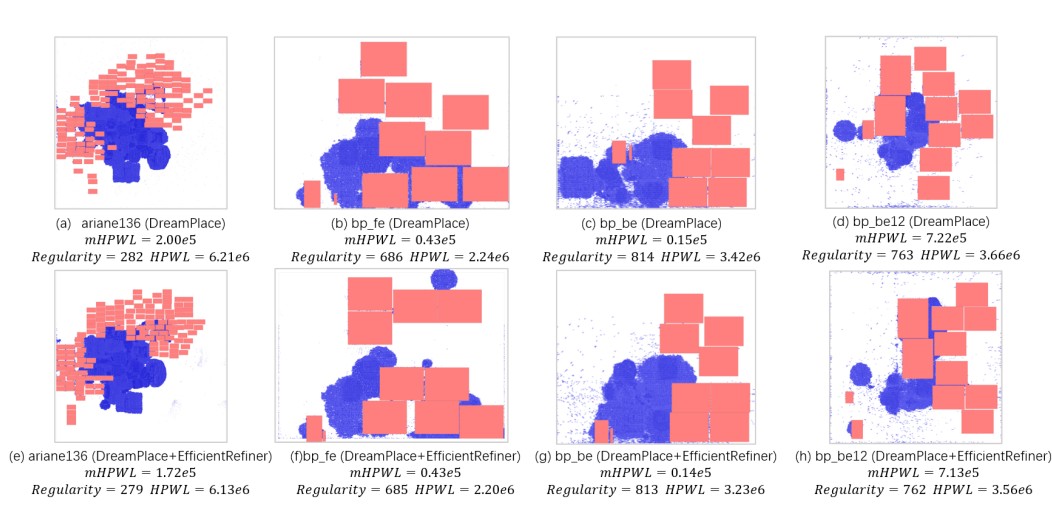

Figure 14: **Layout visualizations before and after refinement for circuits in ChiPBench.**

