# OpenReview forum: "EfficientRefiner: An Efficient Refinement Method over Black-Box Optimization in Macro Placement"
_ICLR.cc/2026/Conference — Submitted to ICLR 2026_

### Official Review · Reviewer_dKCW · 2025-10-25

**Soundness:** 3
**Presentation:** 3
**Contribution:** 2
**Rating:** 2
**Confidence:** 3

**Summary:**

This paper introduces EfficientRefiner, an analytical refinement method designed to enhance macro placements generated by existing Black-Box Optimization (BBO) approaches. A key contribution is a novel fine-grained, module-pair-based overlap formulation that effectively minimizes macro overlaps without causing excessive spreading, which is a limitation of traditional coarse-grained density-based methods in analytical placers. The method is computationally efficient, leveraging GPU acceleration and a pruning strategy to scale to large designs. Experimental results on benchmarks like ISPD2005, ICCAD2015 and ChiPBench demonstrate improvements in Half-Perimeter Wirelength (HPWL) and Power, Performance, and Area (PPA) metrics.

**Strengths:**

1.	The paper is well-structured and clearly written. The proposed EfficientRefiner method demonstrates consistent improvements across multiple benchmarks.
2.	The analytical formulation for macro overlapping, which operates on a fine-grained, module-pair basis, presents a technically sound approach.

**Weaknesses:**

1.	While the idea of using an analytical method specifically for macro refinement is novel, its advantages over established 2-stage mixed-size placers could be more thoroughly justified. The latter inherently considers macro placement with awareness of standard cells, a paradigm well-regarded in the physical design community. The proposed macro-centric refinement, focusing primarily on macro HPWL, might be more susceptible to local optima and yield marginal gains in a full-chip context. The reported significant improvements in mixed-size HPWL are indeed notable but require further explanation to align with the macro-only optimization focus.

2.	As illustrated in Figures 5-7, the regularity metric does not exhibit consistent convergence; for instance, it increases on the superblue1 circuit. This appears counter-intuitive when contrasted with the significant improvement in timing metrics (WNS, TNS) for the same circuit reported in Table 3. The authors should provide an analysis of these observations.

3.	In Table 2, only DREAMPlace is compared on ChiPBench / OpenROAD cases.

4.	There is a potential inconsistency in terminology. The paper positions EfficientRefiner as a refiner for "any Black-Box Optimization (BBO) method," yet a primary baseline used for initialization is DREAMPlace, an analytical placer. Analytical methods like DREAMPlace are typically considered "white-box" due to their explicit, differentiable objective functions.

**Questions:**

1.	To facilitate a comprehensive evaluation and better understand the trade-offs involved, could you provide the results of regularity, macro HPWL, mixed-size HPWL, and PPA metrics in one table?
2.	If permitted by the conference proceedings, including visualizations of the placement results before and after refinement (e.g., for a representative circuit like superblue1) would greatly enhance the intuitive understanding of the improvements achieved by EfficientRefiner.

---

> ### Author Response · Authors · 2025-11-20
>
> **Regularity Trend**
>
> Thank you for the observation. Our code is correct, but when generating Figures 5–7, we accidentally set the Regularity coefficient to 0, which caused the Regularity curves to appear non-convergent. We sincerely apologize for this mistake.
>
> We have fixed the issue in the revised version. The corrected plots show that both HPWL and Regularity decrease during refinement, and decrease rate slows as the iterations proceed.
>
> We have also included the commands for producing the HPWL and Regularity trend, the initial layouts for refinement, and the HPWL/Regularity trends produced by our runs in the "regularity-trend" directory at https://anonymous.4open.science/r/EfficientRefiner-5F66. The results can be reproduced by running the provided scripts.
>
> **Explanation on the term “black-box optimization”**
>
> Thank you for pointing this out. Our original intention in using the term “black-box optimization” was to convey that our method can refine the outputs of any existing placer without relying on its internal mechanics. In that sense, we treated the upstream placer as a black box.
>
> However, we realize that this term could easily be interpreted as referring to specific black-box optimization algorithms, such as simulated annealing, or genetic algorithms. We’re sorry for the confusion, and we will revise the term in the updated version to avoid misunderstanding.
>
> **Layout visualization**
>
> In the revised version (Appendix A.4.7), we provide layout visualizations before and after refinement for both the ICCAD2015 and ChipBench circuits. The results on ChipBench demonstrate reductions in both regularity and macro HPWL. For the ICCAD2015 benchmarks “superblue1” to “superblue3”, our refinement method preserves the general placement patterns produced by DreamPlace while improving both regularity and macro HPWL. The yellow boxes in the figures highlight that our method moves macros closer to the chip boundary, reducing the regularity metric and leaving more whitespace for standard-cell placement. The green boxes show that we also pull tightly connected macros closer together, thereby reducing macro HPWL.
>
> **Insights into why our refinement process can enhance layout quality of analytical methods**
>
> 1. Macro placement has a significant impact on overall placement quality. Even small adjustments to macros can substantially influence the placement of standard cells (as shown in the visualization figures), thereby affecting overall layout metrics. This work follows the established methodology of previous approaches like MaskRegulate, which also employ macro adjustment followed by analytical standard-cell placement to optimize mixed-size layouts.
>
> 2. Lowering regularity (i.e., pulling macros closer to the placement boundary) appears to be an important heuristic for improving mixed-size placement, as also suggested by existing methods like MaskRegulate. Our method also reduces macro HPWL, which contributes to lowering overall HPWL.
> 3. Even when analytical methods employ two-stage optimization, the resulting layouts may still leave room for improvement. The benefits of joint optimization in analytical methods can be largely diminished during legalization. For example, a typical analytical flow is: mixed-size global placement → macro legalization → standard-cell legalization. Since macro overlaps are often significant, the macro legalization step can substantially move the macros, so the original positions from the mixed-size global layout may no longer be applicable. By refining the macro placement from such analytical layouts, our method can further improve overall placement quality.
>
> 4. We adopt a hierarchical approach that first makes fine-grained refinement on macros that have a large influence on the layout (as seen in the visualizations), and then repositions standard cells. This approach can achieve better placement quality compared to analytical methods, which treat all modules equally under a unified loss function.
>
> **Additional refinement results on other baselines**
>
> We selected DreamPlace for refinement, purely macro placement methods generally yield weaker PPA than mixed-size methods based on ChipBench results. We are currently conducting refinement experiments on additional baselines. The PPA results are pending and will be reported upon completion of the time-consuming evaluation process.

---

> ### Author Response · Authors · 2025-11-20
>
> **Table containing Regularity, HPWL and PPA metrics**
>
> The ChipBench evaluation results，including macro HPWL, regularity, mixed-size HPWL, and PPA metrics, are summarized in Table C. When refining the DreamPlace layouts, our method is able to simultaneously reduce macro HPWL and regularity, which in turn improves mixed-size HPWL as well as the WNS and TNS metrics.
>
> Table C: Evaluation of Regularity, HPWL, and PPA Metrics on ChipBench benchmarks. “DP” stands for DreamPlace, and “ER” stands for EfficientRefiner.
>
> | benchmark      | method   | mHPWL  | Regularity    | HPWL      | WNS      | TNS      |
> |---------------|---------|-------|-------|----------|---------|---------|
> | ariane136      | DP      | 200334 | 282.46 | 6211190  | -0.2471 | -208.74 |
> |                | DP+ER   | **172041** | **279.31** | **6133533**  |**-0.2277** | **-166.55** |
> | bp_fe          | DP      | 43468  | 686.23 | 2246648  | -0.6845 | -40.16  |
> |                | DP+ER   | **43020**  | **685.48** | **2204814**  | **-0.3469** | **-19.09**  |
> | bp_be          | DP      | 14997  | 813.50 | 3429613  | -0.6366 | -52.07  |
> |                | DP+ER   | **14251**  | **812.59** | **3230676**  | **-0.6184** | **-49.00**  |
> | bp_be12        | DP      | 722758 | 763.07 | 3659015  | -0.6826 | -65.89  |
> |                | DP+ER   | **713143** | **762.08** | **3560677**  | **-0.6015** | **-54.64**  |
> | bp_multi57     | DP      | 41679  | 348.90 | 6668232  | -2.8632 | -799.80 |
> |                | DP+ER   | **40054**  | **344.54** | **5972371**  | **-2.5053** | **-622.87** |
> | bp68           | DP      |1553817 | 342.27 |12744064  | -2.9514 | -1153.07|
> |                | DP+ER   |**1497953** | **337.59** |**11186402**  |**-2.1447** | **-746.56** |
> | swerv_wrapper  | DP      |223699  | 452.42 | 4642293  | -0.6348 | -543.29 |
> |                | DP+ER   |**206478**  | **451.75** | **4351614**  | **-0.5787** |**-459.99** |
> | VeriGPU        | DP      | **2109**   | 39.13  | 1186895  | -0.5759 | -210.83 |
> |                | DP+ER   | 2452   | **37.72**  | **1174880**  | **-0.3665** | **-66.44**  |

---

> > ### Comment · Reviewer_dKCW · 2025-11-26
> >
> > Thank you for your response, which has addressed some of my concerns. However, most of the concerns stated in Weakness still remain. I will keep my score. See the following text for a point-to-point explanation.
> >
> > 1. The two-stage analytical placer first fixes the macros after the initial mixed-size placement, then repositions the standard cells based on the fixed macro locations [1]. This also allows reoptimization of cell positions around the perturbed macros. To the best of my knowledge and experience, implementing DREAMPlace with this two-stage flow can yield significantly better performance on the ICCAD 2015 benchmarks than the results you reported. Could you clarify whether your DREAMPlace results were obtained using the two-stage approach?
> >
> > [1] Stronger Mixed-Size Placement Backbone Considering Second-Order Information.
> >
> > 2. While regularity does converge, the improvement is marginal. As shown in Appendix A.4.7, there is no discernible pattern in peripheral regularity.
> >
> > 3. Additional comparisons on ChiPBench cases are still lacking. At a minimum, MaskRegulate should be included for comparison, as was done in Table 3.
> >
> > 4. Black-box optimization (BBO) is a well-defined concept and should not be reinterpreted loosely to suit a particular narrative. Given that your paper’s title includes the term BBO, the content should align closely with the domain of black-box optimization. However, your proposed method and experiments do not employ any BBO techniques; rather, they rely primarily on white-box analytical approaches. This, in my view, constitutes a fundamental conceptual discrepancy.

---

> ### Author Response · Authors · 2025-11-26
>
> Thank you very much for your thoughtful comments. Our responses are as follows:
>
> 1.	We have reviewed reference [1], but we did not find reported results on the ICCAD 2015 benchmarks in that paper.
>
>      In our manuscript, we have compared against the method in [1], which we refer to as DreamPlace 4.1.0. The DreamPlace 4.1.0 results on ICCAD 2015 are reported in Table 8 of the appendix. We have also reported refinement results on DreamPlace 4.0 for the same benchmark in Table 3 of the main paper. As expected, the DreamPlace 4.1.0 results in Table 8 are better than the DreamPlace 4.0 results in Table 2.
>
>     Given this, we wonder whether your impression that “DreamPlace performs much better than what we reported” might come from looking at the DreamPlace 4.0 results in Table 2 rather than the DreamPlace 4.1.0 results in Table 8.
>
>     If this is not the case, we would greatly appreciate it if you could share references that report DreamPlace performance on the ICCAD2015 benchmarks. Such information would be very valuable for us in further evaluating and comparing our method.
>
> 2.	Regarding regularity, we think: (1) Regularity is one component of our loss function. As the results shown, regularity does decrease in our experiments. (2) Regularity is a proxy metric, while the actual measure of final placement quality is PPA. Therefore, PPA is the more meaningful indicator to focus on. Although the improvement in regularity is relatively small, our method achieves notable gains in the final PPA metrics, as reported in Table 2 of the paper.
>
> 3.	We are currently evaluating the PPA metrics of additional methods including MaskPlace, WireMask-EA, EfficientPlace, and ChipFormer on the ChiPBench cases, and we will report the results once these experiments are completed. Regarding MaskRegulate, its public implementation supports the ICCAD 2015 benchmarks but does not support the ChiPBench designs for several reasons:
> (1)	The regular-expression–based parsing in the code cannot correctly handle the ChiPBench file formats.
> (2)	ChiPBench hase multiple .lef and .lib files, whereas the current implementations of MaskRegulate and DreamPlace support only a single file of each type.
> (3)	The module names in ChiPBench cannot be correctly parsed by MaskRegulate.
> Due to the substantial engineering effort required to support ChiPBench, currently we have not included MaskRegulate in our experiments.
>
> 4.	Thank you for pointing out the issue regarding the term "BBO." Following your earlier comment, we have updated the term “BBO” throughout the revised manuscript and have also revised the title. The updated version of the manuscript has already been uploaded. The current title is: “EfficientRefiner: An Efficient Refinement Method for Macro Placements Generated by Off-the-shelf Placers.” However, we currently do not have the permission to update the title shown on OpenReview, so the title displayed in the system remains the previous version.

---

> > ### Author Response · Authors · 2025-12-03
> >
> > We conducted additional experiments on the OpenROAD “bp68” circuit to evaluate our refinement procedure on layouts produced by other placement methods. The results are shown in the Table D. Our refinement improves the layout from both DreamPlace 4.1.0 and EfficientPlace, improving WNS by 27% and 7.0% and improving TNS by 35.3% and 12.6%, respectively. For WireMask-EA, the OpenROAD evaluation failed at the detailed placement stage. We suspect that this may occur because the macros were too densely packed. After applying our refinement procedure, where the Regularity metric guide some macros towards the boundary, the flow completed successfully. This may also indicate that the refined layout is better than the original one.
> >
> > In addition, the DreamPlace 4.1.0 results reported in our paper achieve the best performance among all baseline methods in Table D. After applying EfficientRefiner, DreamPlace 4.1.0 attains the best overall results. These findings demonstrate that the baseline method chosen in our paper is strong and that it can still achieve further improvement after refinement.
> >
> > Table D: Refinement results for different baselines on “bp68” circuit.  “ER” is short for EfficientRefiner.
> > | Method               | HPWL(↓)   | WL(↓)      | Cong(↓)  | Power(↓) | NVP(↓) | WNS(↑)    | TNS(↑)      | Area(↓) |
> > |----------------------|-----------|------------|----------|----------|--------|-----------|-------------|---------|
> > | DreamPlace4.1.0      | 12744064  | 14728606   | 0.4597   | 0.1530   | 2427   | -2.9514   | -1153.07    | 275709  |
> > | DreamPlace4.1.0+ER   | **11186402**  | **12856599**   | **0.4037**   | **0.1485**   | **563**    | **-2.1447**   | **-746.56**     | **269561**  |
> > | | | | | | | | | |
> > | EfficientPlace       | 20555495  | **23466686**   | **0.7304**   | **0.1596**   |**1117**   | -3.5916   | -1802.31    | 316269  |
> > | EfficientPlace+ER    | **19628218**  | 23819069   | 0.7387   | 0.1639   | 1126   | **-3.3406**   | **-1575.43**    | **294475**  |
> > | | | | | | | | | |
> > | WireMask-EA          | Detailed placement failed | —— | —— | —— | —— | —— | —— | —— |
> > | WireMask-EA+ER       | **19093276**  | **21671670**   | **0.6787**   | **0.1565**   | **1396**   | **-2.6638**   | **-1685.01**    | **307166**  |

---

### Official Review · Reviewer_TgZh · 2025-10-30

**Soundness:** 3
**Presentation:** 2
**Contribution:** 3
**Rating:** 6
**Confidence:** 3

**Summary:**

This paper proposes a novel refinement method for the layout outputs of macro placement, an essential procedure of the chip placement. Compared to traditional analytical methods (quantifying the density but cannot prevent macro overlapping) and RL-based methods (using position mask to indicate the overlap location), this paper proposes to directly minimize the overlap areas. Furthermore, it proposes an acceleration mechanism for calculation. Experimental metrics (HPWL, PPA metrics) clearly show the efficiency of EfficientRefiner.

**Strengths:**

- The idea of directly optimizing the overlap areas is novel and interesting. The areas objective can be viewed as a relaxation of the overlap, and experiments in Appendix A.4.6 show that EfficientRefiner can also prevent macro overlap as the refinement process progresses.
- The efficient calculation and optimization of the proposed objective is reasonable, and the experiments show the efficiency in both refinement performance and operation time.
- The experimental evaluation seems solid, showing improvements in both wirelength and PPA metrics.

**Weaknesses:**

- I don’t understand why the manuscript claims that EfficientRefiner is a refinement method over black-box optimization methods. It can also be applied to the final layouts of other methods like RL-based methods (MaskPlace and Chipformer in your experiments) or analytical methods (DreamPlace in your experiments), isn’t that?
- Typos:
    - line 016: “degradation resulted from” → “degradation resulting from”;
    - line 049: “overlaps which has to be” → “overlaps which have to be”;
    - line 113: “ChipFormer, improve” → “ChipFormer improves”;
    - line 115: “the most effective RL method require” → “the most effective RL method requires”;
    - line 121: “through numerous adjustment” → “through numerous adjustments”;
    - line 123: “LaMPlace … but guide” → “LaMPlace … but guides”;
    - line 136: “learns a adjustment policy” → “learns an adjustment policy”;
    - line 153: “The optimization objective include” → “The optimization objectives include”;
    - line 156: “smaller HPWL may indicates” → “smaller HPWL may indicate”;
    - line 170: “EfficientRefiner first represent” → “EfficientRefiner first represents”;
    - line 175: “optimizes a joint objective function consist of” → “optimizes a joint objective function consisting of”;
    - The term *DreamPlace* is not unified over the manuscript, while in Section 2.2, it is *Dreamplace*.
    - line 196: “it refine the” → “it refines the”;
    - line 208: “along with prunning” → “along with pruning”;
    - line 400: “efore and after” → “before and after”;
    - line 452: “Effectiveness of the Fine-grained Overlap Modeling” → “Effectiveness of the Fine-grained Overlap Modeling.”;
    - line 463: “Parameter Analysis” → “Parameter Analysis.”;
    - line 477: “it accelerate” → “it accelerates”;
    - line 729: “Return: $ \hat{Overlap} _ {xij} $, $\hat{Overlap} _ {xij}$” → “Return: $\hat{Overlap} _ {xij}$, $\hat{Overlap} _ {yij}$”;
    - line 755: “Return: $\delta\hat{Overlap} _ {xij}$, $\delta\hat{Overlap} _ {xij}$” → “Return: $\delta\hat{Overlap} _ {xij}$, $\delta\hat{Overlap} _ {yij}$”;
    - line 817: “8192 models” → “8192 modules” ;
    - line 885: “not accessible fot us” → “not accessible for us”;
    - line 892: “the results shows” → “the results show”.

**Questions:**

- In Eq. (6), the condition should be $|x_i-x_j|<\frac{w_i}{2} + \frac{w_j}{2}$, instead of $|x_i-x_j|<\frac{w_j}{2} + \frac{w_j}{2}$?
- In line 1052, you mentioned that “Fig.8-11 show the overlap growth before and after legalization”. It should be HPWL growth?
- Based on my understanding, from the illustration in the manuscript, the reference of LaMPlace seems wrong. Do you aim to cite [1] instead?

## References

[1] LaMPlace: Learning to Optimize Cross-Stage Metrics in Macro Placement. ICLR 2025.

---

> ### Author Response · Authors · 2025-11-20
>
> **Explanation on the term “black-box optimization”**
>
> Our method can indeed refine placements produced by RL-based and analytical placers.
>
> Our original intention in using the term “black-box optimization” was to convey that our method can refine the outputs of existing placer without relying on its internal mechanics. In that sense, we treat the upstream placer as a black box.
>
> Thank you for pointing this out. We recognize that this term may be misunderstood as referring to specific black-box optimization algorithms, such as simulated annealing or genetic algorithms. We apologize for the confusion, and we have revised the term in the updated version.
>
> **Typos and Citation Issues**
>
> Thank you for pointing out the typos and the errors in the descriptions related to Equation (6) and Figures 8–11. We have corrected these issues in the revised version.
>
> LaMPlace refers to the work cited as [1]. We have corrected the citation in the updated manuscript.

---

> > ### Comment · Reviewer_TgZh · 2025-11-25
> >
> > Thanks for your reply. My concerns are addressed. After reading other reviewers' comments and the authors' updates of the manuscript, I think this paper is of high quality with principled methodology and high-efficient GPU acceleration implementation. The refinement effectiveness compared to other refinement methods is also impressive. Thus, I'll raise my ratings.

---

> > > ### Author Response · Authors · 2025-11-25
> > >
> > > We sincerely appreciate your time and thoughtful feedback. Your comments greatly helped us improve the clarity of the manuscript. We are also very grateful for your positive assessment of our methodology, GPU-accelerated implementation, and the effectiveness of our refinement approach compared with existing methods. Thank you for your encouraging follow-up.

---

### Official Review · Reviewer_4rZs · 2025-10-31

**Soundness:** 3
**Presentation:** 3
**Contribution:** 2
**Rating:** 4
**Confidence:** 4

**Summary:**

This paper introduces EfficientRefiner, a gradient-based refinement method that improves macro placements produced by any black-box optimizer. It introduces a pairwise overlap function that exactly measures macro-to-macro overlaps and yields non-zero gradients. Also a boundary-oblivious encoding that maps legal positions to unconstrained learnable vectors. So optimization is box-free until a final legalization step. EfficientRefiner drops HPWL by 7%-35%, boosts WNS by 20% and TNS by 29%. While running <=10 mins outperforming recent RL refiners without any training tuning.

**Strengths:**

- EfficientRefiner replaces coarse grid-density penalties with an exact macro-pair overlap function whose gradients stay non-zero until every overlap vanishes, so legalization needs <1 % HPWL movement instead of 8–60 %.
- GPU-parallel overlap evaluation plus bin-pruning cuts runtime 1000× versus naive loops, refining 8 k macros in ~130 s—no training, no placer-specific tuning.
- It plugs as a black-box post-process to any placer, delivering 7–35 % HPWL savings and 20 % WNS / 29 % TNS gains on standard benchmarks with stable, dataset-independent hyper-parameters.

**Weaknesses:**

EfficientRefiner currently focuses only on macro placement and treats standard cells as fixed, so it cannot simultaneously optimize mixed-size layouts and may miss macro-cell coupling opportunities. Its pairwise overlap model, although pruned, still scales quadratically with macro count, risking memory and compute growth for future designs with tens of thousands of macros. Finally, the method relies on simplified HPWL and regularity proxies rather than true routed timing or congestion metrics, which could limit the correlation with final PPA on advanced nodes.

**Questions:**

- How does the pairwise overlap formulation handle macros of vastly different sizes (e.g., a large memory block next to a small IP) without introducing imbalance in gradient magnitudes?
- The boundary-oblivious representation maps legal positions to unbounded learnable vectors. How sensitive is the optimization trajectory to the initial values of these vectors, and could a poor initialization trap the solver in a high-overlap region?
- The overlap weight α is set to 10^5 for all experiments. Is there a systematic way to choose α based on design characteristics (utilization, macro count, aspect ratio) rather than relying on a single fixed value?
- While pruning reduces quadratic growth, the worst-case memory still scales as O(n²). What is the largest macro count the authors have tested, and what are the projected bottlenecks for 100 k-macros designs?
- All reported PPA numbers are either pre-route estimates (OpenTimer) or surrogate metrics (HPWL, regularity). Has the team run any confidential full-flow experiments to verify that the 20 % WNS and 29 % TNS gains survive detailed routing and clock-tree synthesis?

---

> ### Author Response · Authors · 2025-11-20
>
> **Scalability**
>
> 1. Our method requires O(n) memory rather than O(n^2). The reason is that we do not store the overlap gradients for all module pairs. Once the gradient for a pair is computed, it is immediately accumulated into the corresponding module’s total gradient.
>
> 2. Although the theoretical worst case time complexity is O(n^2). We partition the placement region into bins and compute overlap only between a module and the modules located in the same or neighboring bins. This reduces computation by a large constant factor. For example, in our experiments on designs with several hundred thousand modules（described in the next paragraph）, the number of bins is more than 300^2. Since refinement starts from an existing placement, modules are approximately uniformly distributed. Under this distribution, restricting overlap evaluation to the same and adjacent bins effectively reduces the work by a factor of about 300^2/9=10k, which substantially improves efficiency.
>
> 3. Our method remains practical for cases with extremely large number of modules. We have evaluated refinement on **all macros and standard cells** on the ISPD2005 circuits “adaptec1”–“adaptec4,” where the total number of modules ranges from **211k to 496k**. Table A (rows 2–3) reports HPWL before and after refinement. The initial layouts are generated by DreamPlace 4.0, and EfficientRefiner runs on an NVIDIA RTX 3090Ti GPU (24GB memory) with 5k refinement iterations. As shown, EfficientRefiner can handle circuits with several hundred thousand objects within reasonable memory and runtime budgets.
>
> Table A: HPWL (×1e7) and runtime for different refinement strategies on the ISPD2005 circuits. “DP” stands for DreamPlace, and “ER” stands for EfficientRefiner.
> | Method| adaptec1| adaptec2| adaptec3| adaptec4|
> |-|-|-|-|-|
> | DP| 9.28| 12.39| 18.15 | 21.85|
> | DP + ER on mixed-size placement| _8.46_ (0.41h) | _12.09_ (0.52h) | _16.21_ (0.98h) | _20.88_ (1.08h) |
> | DP + ER on macros + DP for standard cells layout | **6.88** (42s)  | **7.93** (43s)  | **13.82** (46s) | **18.53** (59s) |
>
> We would also like to emphasize that for designs with very large number of modules, the macro-first refinement strategy adopted in our paper produces better results. Instead of refining all modules jointly, we can first focus on the large macros using EfficientRefiner, and then place remaining standard cells with DreamPlace. This hybrid approach is substantially more efficient than joint refinement and achieves larger HPWL reductions (as shown in rows 3-4 in Table A). Based on these results, for hypothetical scenarios involving up to 100k macros, we recommend selectively refining only the large modules to obtain greater improvements with far lower computational cost.
>
> 4. Macros refer to large functional units such as SRAMs, ROMs, and major IP blocks. To our knowledge, it is highly unlikely to have near 100k macros in practical industrial designs.
>
> **Clarification on PPA results**
>
> We would like to clarify that the PPA results reported for the ChiPBench circuits (Table 2) are obtained using the open-source OpenROAD toolchain which runs the **complete placement flow** including clock tree synthesis and routing. These results are **not** pre-route estimates. The reported 20% WNS and 29% TNS improvements refer to these **full-flow outcomes**.
> For the ICCAD2015 benchmark, as OpenROAD does not fully support these designs and commercial tools are currently not accessible to us, we report pre-route estimates with OpenTimer instead.
>
> **Sensitivity to initail placement**
>
> In the boundary-oblivious representation, the Sigmoid function is used to map legal positions to unbounded vectors. Since the Sigmoid has small gradients near the boundaries of the placement region, modules located very close to the boundary may have small gradients. In extreme cases where many modules are densely packed along the boundary, module movement could become difficult. However, we apply our method to refine layouts produced by existing placers, and such highly boundary-crowded patterns are rare in layouts generated by those baselines. In our experiments, all the initial layouts produced by the methods listed in Table 1 can be effectively refined using our approach.
>
> **Addressing modules with Different sizes**
>
> According to Eq. (9) in the manuscript, when a large macro and a small macro overlap, the overlap gradients of both macros have equal magnitude. Thus, we do not think the formulation introduce gradient imbalance.

---

> ### Author Response · Authors · 2025-11-20
>
> **Current focus on macro placement**
>
> Our current work indeed focuses primarily on refining macro placements. Our view is as follows:
>
> 1. Macros have large influence on the overall placement quality due to their large area and high connectivity. Many recent works, such as MaskRegulate and LaMPlace, focus primarily on macro placement and delegate standard-cell placement to analytical tools, and still achieve strong mixed-size HPWL and PPA performance. Our approach follows the similar rationale.
>
> 2. We have also experimented with applying EfficientRefiner to full mixed-size placements (as shown in Table A. The results indicate that refining only the macros, while relying on analytical placers to handle the standard-cell placement, yields better overall quality and significantly lower computational cost compared to jointly refining both macros and standard cells. Therefore, considering both efficiency and effectiveness, our current refinement strategy focuses primarily on macros.
>
> **Current focus on proxy metrics**
>
> Thanks for your suggestions. Our view is as follows:
>
> 1. Routing is extremely time-consuming, which makes repeated post-routing evaluations computationally prohibitive during refinement. Therefore, we currently rely on proxy objectives as in existing methods. In fact, optimizing these proxy metrics already yields significant improvements in the final post-routing PPA results according to our experiments.
>
> 2. Our framework is also flexible to accommodate alternative objectives, including more accurate predictors of post-routing metrics. For example, LaMPlace fits a polynomial model to estimate PPA metrics. Such predictors could be incorporated into our optimization. Since the publicly released data for that work is not yet complete, we leave this integration to future work.
>
> **Choice of overlap weight**
>
> As shown in the parameter sensitivity analysis for macro layouts (Appendix A.4.6) and for mixed-size layouts (Table B), once $\alpha > 1000$, both macro HPWL and mixed-size HPWL vary slightly with further increases in $\alpha$, and the negative impact of legalization also remains small. This suggests that when $\alpha$ is sufficiently large, its value has only small influence on the final placement quality. We therefore set $\alpha$ to a large value.
>
> We believe that automatically determining an appropriate value of $\alpha$ is an important direction. We appreciate your suggestion and will explore it in future work.
>
> Table B: Effect of $\alpha$ on Mixed-Size Layout Refinement
>
> | $\alpha$    | normalized macroHPWL (pre-legalization) |  normalized macroHPWL (post-legalization) | Regularity (pre-legalization) | Regularity (post-legalization) | overlap rate(%) | mixed-size HPWL(x1e7) |
> |------|----------------|-----------------|------------|------------|---------|------------|
> | 0    | 39.99          | 199.74          | 21.99      | 191.17     | 261.54  | 8.22       |
> | 10   | 40.02          | 210.51          | 28.51      | 195.10     | 66.58   | 8.54       |
> | 100  | 59.11          | 165.88          | 58.48      | 180.16     | 17.06   | 7.56       |
> | 1k   | 101.85         | 106.16          | 153.27     | 193.19     | 0.09    | 6.92       |
> | 5k   | 101.36         | 103.97          | 183.56     | 205.95     | 0.00    | 6.72       |
> | 10k  | 101.49         | 104.74          | 196.03     | 210.98     | 0.00    | 6.74       |
> | 50k  | 104.18         | 105.25          | 221.64     | 229.63     | 0.00    | 6.99       |
> | 100k | 108.04         | 108.69          | 228.86     | 233.79     | 0.00    | 6.92       |

---

### Official Review · Reviewer_pvFu · 2025-11-03

**Soundness:** 4
**Presentation:** 2
**Contribution:** 4
**Rating:** 8
**Confidence:** 5

**Summary:**

This paper focuses on the macro placement refinement stage in EDA, addressing two key issues in state-of-the-art methods: RL-based approaches lack full-layout awareness, leading to suboptimal results, while analytical methods suffer quality degradation from overlap-resolving legalization. The goal is to reduce computational overhead in the refinement stage while improving layout quality, compensating for the limitations of existing BBO placement methods. The proposed EfficientRefiner uses an analytical framework that encodes macro positions as unbounded learnable vectors, optimizes an objective integrating target metrics (HPWL, regularity) and a novel fine-grained module-pair-based overlap function, and enhances efficiency through pruning algorithms and GPU acceleration. Experiments on ISPD2005, ICCAD2015, and ChipBench benchmarks show that EfficientRefiner significantly improves PPA metrics.

**Strengths:**

1. Comprehensive background and related work.
2. Novel fine-grained overlap formulation solves critical limitations of existing methods.
3. High efficiency and scalability via GPU acceleration.
4. EfficientRefiner seamlessly integrates with RL-based (MaskPlace, Chipformer), BBO (WireMask-EA), and analytical (DreamPlace, NTUPlace3) placement methods, demonstrating consistent improvements across most baselines.

**Weaknesses:**

1. I recommend providing detailed analysis of key parameter sensitivity, such as how to adapt $\alpha$ for different cases with various scales, as well as the influence of bin size.
2. The presentation should be improved. For example, please consider moving some information in Figure 1 to the wrapped figure embedded in the main context for easier understanding.
3. (Minor) I appreciate the efficiency of the proposed method and believe it is a remarkable advantage. The corresponding analysis should be included in the main text.
4. (Minor) Use \mathrm{} for the text in equations, such as Overlap and HPWL.

**Questions:**

The paper acknowledges that ICCAD2015 PPA results are pre-routing estimates using OpenTimer (not including routing effects), and commercial tools are unavailable for accurate evaluation. Could you analyze the correlation between pre- and post-routing results? For example, analyze the cases from ChipBench.

---

> ### Author Response · Authors · 2025-11-20
>
> **Choice of Parameters**
>
> We choose the current value of $\alpha$ based on parameter sensitivity analysis for macro layouts (Appendix A.4.6) and for mixed-size layouts (Table B), once $\alpha > 1000$, both macro HPWL and mixed-size HPWL vary slightly with further increases in $\alpha$, and the negative impact of legalization also remains small. This suggests that when $\alpha$ is sufficiently large, its value has only small influence on the final placement quality. We therefore set $\alpha$ to a large value. We will further explore how to automatically determine an appropriate $\alpha$ in future work.
>
> The underlying rationale for selecting the bin size is as follows. For designs containing a very large number of modules, choosing a bin size that is slightly larger than most small modules helps reduce the number of module pairs to be computed. In the datasets, module sizes vary significantly. For example, in the mixed-size refinement experiments on the ISPD2005 benchmarks (Table A), standard cells are much smaller than macros. The bin size can be set slightly larger than the maximum size of all standard cells. However, if the bin size is set too small, resulting in a very large number of bins, memory consumption may exceed the GPU’s capacity. Therefore, we determine the lower bound of bin size according to the available GPU memory in practice.
>
> Table B: Effect of $\alpha$ on Mixed-Size Layout Refinement
>
> | $\alpha$    | normalized macroHPWL (pre-legalization) |  normalized macroHPWL (post-legalization) | Regularity (pre-legalization) | Regularity (post-legalization) | overlap rate(%) | mixed-size HPWL(x1e7) |
> |------|----------------|-----------------|------------|------------|---------|------------|
> | 0    | 39.99          | 199.74          | 21.99      | 191.17     | 261.54  | 8.22       |
> | 10   | 40.02          | 210.51          | 28.51      | 195.10     | 66.58   | 8.54       |
> | 100  | 59.11          | 165.88          | 58.48      | 180.16     | 17.06   | 7.56       |
> | 1k   | 101.85         | 106.16          | 153.27     | 193.19     | 0.09    | 6.92       |
> | 5k   | 101.36         | 103.97          | 183.56     | 205.95     | 0.00    | 6.72       |
> | 10k  | 101.49         | 104.74          | 196.03     | 210.98     | 0.00    | 6.74       |
> | 50k  | 104.18         | 105.25          | 221.64     | 229.63     | 0.00    | 6.99       |
> | 100k | 108.04         | 108.69          | 228.86     | 233.79     | 0.00    | 6.92       |
>
> **Correlation Analysis between Pre- and Post-Routing Results**
>
> We appreciate your suggestion, and we agree that analyzing the correlation between pre- and post-routing PPA would be highly valuable. We attempted to evaluate the pre-routing results of the ChipBench circuits using the same DreamPlace+OpenTimer flow applied to the ICCAD2015 benchmarks. However, the current evaluation tool does not support the multiple .lib files required by the ChipBench designs and cannot correctly parse certain module-name formats in these circuits. We will continue exploring evaluation approaches that can properly support the ChipBench layouts in future work.
>
> **Organization and Writing Issues**
>
> Thank you for your helpful suggestions. We have updated the formatting of the in-text equations accordingly. Considering the page limit, we will restructure Figure 1 and include the efficiency analysis in the main text in the final revised version.

---

### Meta-Review · Area_Chair_gNSn · 2026-01-07

**Summary:**

The paper proposes EfficientRefiner, an efficient analytical refinement framework for improving macro placements produced by off-the-shelf placers via gradient-based optimization with a novel fine-grained pairwise overlap formulation and GPU-accelerated pruning. The majority of reviewers find the approach technically sound and practically useful, and the rebuttal resolves most concerns. However, one reviewer maintains strong objections—questioning whether the evaluation uses the strongest two-stage mixed-size DreamPlace setting, noting that regularity improvements appear marginal and insufficiently explained relative to the reported PPA gains, arguing that baseline/full-flow comparisons on benchmark. Considering these issues, I recommend rejecting the paper and strongly encourage the authors to revise it based on the review comments for resubmission.

**Reviewer Concerns:**

All concerns raised by Reviewer TgZh have been addressed (and the reviewer raised their score). Most concerns raised by Reviewer pvFu and Reviewer 4rZs were addressed. The key concerns raised by Reviewer dKCW regarding framing/positioning and baseline/comparison completeness are still  outstanding.

**Reviewer Scores:**

I think the final scores of the reviewers are 8 (pvFu), 4 (4rZs), 8 (TgZh), 2 (dKCW).

---

### Decision · Program_Chairs · 2026-01-26

Reject